# Phenotypic signatures of immune selection in HIV-1 reservoir cells

Weiwei Sun[1,2,7], Ce Gao[1,2,7], Ciputra Adijaya Hartana[1], Matthew R. Osborn[1], Kevin B. Einkauf[1,2], Xiaodong Lian[1,2], Benjamin Bone[1,2], Nathalie Bonheur[1], Tae-Wook Chun[3], Eric S. Rosenberg[4], Bruce D. Walker[1,4,5,6], Xu G. Yu[1,2,8] & Mathias Lichterfeld[1,2,8 ✉]

Human immunodeficiency virus 1 (HIV-1) reservoir cells persist lifelong despite antiretroviral treatment[1,2] but may be vulnerable to host immune responses that could be exploited in strategies to cure HIV-1. Here we used a single-cell, next-generation sequencing approach for the direct ex vivo phenotypic profiling of individual HIV-1-infected memory CD4+ T cells from peripheral blood and lymph nodes of people living with HIV-1 and receiving antiretroviral treatment for approximately 10 years. We demonstrate that in peripheral blood, cells harbouring genome-intact proviruses and large clones of virally infected cells frequently express ensemble signatures of surface markers conferring increased resistance to immune-mediated killing by cytotoxic T and natural killer cells, paired with elevated levels of expression of immune checkpoint markers likely to limit proviral gene transcription; this phenotypic profile might reduce HIV-1 reservoir cell exposure to and killing by cellular host immune responses. Viral reservoir cells harbouring intact HIV-1 from lymph nodes exhibited a phenotypic signature primarily characterized by upregulation of surface markers promoting cell survival, including CD44, CD28, CD127 and the IL-21 receptor. Together, these results suggest compartmentalized phenotypic signatures of immune selection in HIV-1 reservoir cells, implying that only small subsets of infected cells with optimal adaptation to their anatomical immune microenvironment are able to survive during long-term antiretroviral treatment. The identification of phenotypic markers distinguishing viral reservoir cells may inform future approaches for strategies to cure and eradicate HIV-1.

At present, a cure for HIV-1 infection is considered elusive owing to infected cells that harbour genome-intact, chromosomally integrated viral DNA and persist long-term despite suppressive antiretroviral treatment (ART)[1,2]. After initiation of ART, the frequency of these cells, here termed HIV-1 reservoir cells, declines over time[3]; however, this process is slow, and the mechanisms underlying this decline are not well understood. ART itself is unlikely to directly contribute to the longitudinal decrease of viral reservoir cells because antiviral drugs have no cytotoxic activity against virally infected cells; they merely protect HIV-uninfected cells against viral infection and prevent the seeding of new viral reservoir cells. Instead, it is likely that immune mechanisms have an important role in the longitudinal reduction of HIV-1 reservoir cells during ART, specifically for the decline of the small subset of cells encoding for functional, genome-intact proviral sequences, which seem to decrease faster compared to defective proviruses[4–7]. Footprints of antiviral immune effects against the HIV-1 reservoir cells may be more visible when qualitative features of HIV-1-infected cells, including proviral chromosomal integration sites and transcriptional activities, are analysed[8–12], and when viral reservoir profiles are evaluated over

extended periods of suppressive ART. However, previous studies suggest that viral reservoir cells can frequently resist host selection forces, and long-term persistence of highly transcriptionally active proviruses that seem to effectively avoid host immune activity has been reported in multiple studies[10,13,14]. Mechanisms that allow HIV-1-infected cells to withstand host immune effects and to survive lifelong are not well defined at present.

## Single-cell proteogenomic profiling

Interrogation of the surface phenotype of HIV-1 reservoir cells may be highly informative for understanding the susceptibility or resistance of HIV-1 reservoir cells to host immune responses; however, such investigations have been precluded in the past by technical limitations that did not allow profiling the surface phenotype of these cells directly ex vivo, and instead relied on a phenotypic analysis of a small subset of reservoir cells that produce viral antigen in response to in vitro stimulation[15,16] or on a biocomputational inference of the reservoir cell profile[17]. As an alternative approach, we developed an experimental

[1]Ragon Institute of MGH, MIT and Harvard, Cambridge, MA, USA. [2]Infectious Disease Division, Brigham and Women's Hospital, Boston, MA, USA. [3]National Institute of Allergies and Infectious Diseases, Bethesda, MD, USA. [4]Infectious Disease Division, Massachusetts General Hospital, Boston, MA, USA. [5]Howard Hughes Medical Institute, Chevy Chase, MD, USA. [6]Institute for Medical Engineering and Sciences and Department of Biology, Massachusetts Institute of Technology, Cambridge, MA, USA. [7]These authors contributed equally: Weiwei Sun, Ce Gao. [8]These authors jointly supervised this work: Xu G. Yu, Mathias Lichterfeld. ✉e-mail: mlichterfeld@partners.org

strategy, here termed 'phenotypic and proviral sequencing' (PheP-seq), which was designed to evaluate the phenotype of patient-derived HIV-1-infected cells, using a single-cell next-generation sequencing assay[18] (Extended Data Fig. 1a) that permits one to jointly profile phenotypic markers and selected parts of chromosomal DNA from single cells[19]. This approach complements recent studies using cellular indexing of transcriptomes and epitopes by sequencing (CITE-seq) that focus on a combined analysis of phenotypic markers and the cellular transcriptome in HIV-1 reservoir cells[20,21]. For our experiments, memory CD4+ (mCD4+) T cells isolated from the peripheral blood (PB) of study participants infected with HIV-1 (n = 5) were labelled with a cocktail of oligonucleotide-tagged antibodies directed to T cell surface markers (n = 53), without any type of prior in vitro activation or manipulation. Afterwards, cells were subjected to a single-cell analytic platform technique designed to conduct a multiplex polymerase chain reaction (PCR) for amplifying small fragments of genomic HIV-1 DNA, coupled with amplification of corresponding antibody-bound oligonucleotide tags. Amplification products were then pooled, sequenced and biocomputationally deconvoluted to isolate sequencing reads originating from individual cells. For a deep interrogation of the proviral sequence in each infected cell, a total of 18 non-overlapping custom-designed primer pairs were synthesized that spanned strategically important and phylogenetically conserved regions in the HIV-1 proviral genome and allowed for simultaneous amplifications of small HIV-1 DNA fragments approximately 200–300 base pairs (bp) in length from single virally infected cells (Fig. 1a,b, Extended Data Fig. 1b and Supplementary Table 1); specifically, we included primers from the intact proviral DNA assay (IPDA)[22] that permit identifying proviruses with high probability to be genome-intact. For enhanced analytic depth, we also included primers spanning the virus–host chromosomal integration site junctions of large, sequence-identical proviral clones previously characterized in our study participants; amplification of such previously defined chromosomal regions allows one to unequivocally identify clonal HIV-1-infected cells with known proviral sequences and to relate phenotypic information of infected cells to proviral chromosomal locations.

## Global analysis of HIV-1 reservoir cells

Using this assay, we analysed 530,143 individual mCD4+ T cells in PB from five study participants (four who had remained on suppressive ART for approximately 10 years, and one maintaining undetectable levels of HIV-1 plasma viraemia in the absence of ART (Extended Data Fig. 2a,b). To analyse the proviral landscapes in infected cells, we introduced the following classification system (Fig. 1b,c, Extended Data Figs. 1b and 2c and Supplementary Table 2). Category 1 included cells (n = 2,859) considered HIV-1-infected and harbouring any type of HIV-1 proviruses; for added rigour, we required these cells to have a total of at least 20 proviral reads from at least two HIV-1 amplicons. Category 2 contained cells (n = 193) harbouring proviruses enriched for genome-intact HIV-1 sequences[23,24] and were defined by having a total of at least 20 sequencing reads from the two IPDA amplicons. In case an IPDA amplicon was missing (owing to, for example, suboptimal PCR performance), a total of at least 15 proviral amplicons had to be present to qualify for inclusion in category 2. Sequences showing statistically significant signs of hypermutation mediated by APOBEC3G or APOBEC3F were excluded from category 2. Category 3 included cells (n = 125) belonging to large clones of virally infected cells that were defined by amplicons spanning proviral integration sites (Extended Data Fig. 1c) or, in a limited number (n = 32) of cells, by completely identical proviral sequences from all available amplicons (Fig. 1d). This category was further subdivided into cells containing genome-intact clonal proviruses (category 3.1, n = 56 cells) and defective clonal proviruses (category 3.2, n = 69 cells), and exhibited expected clonal phylogenetic clusters (Fig. 1d and Extended Data Fig. 2c). Category 4 (total of

n = 398) cells harboured proviral genomes with statistically significant sequence hypermutations. For comparative purposes, cells without detectable proviral sequencing reads (category 0) were considered as HIV-uninfected cells.

For an initial global analysis of the phenotype of HIV-1-infected cells from PB, we visualized in silico-gated CD3+CD4+ cells (after exclusion of contaminating CD45RA+CCR7+ naive T cells) from the different categories on uniform manifold approximation and projection (UMAP)[25] plots, classifying the mCD4+ T cell pool in five distinct, computationally defined phenotypic clusters (Fig. 2a,b). This analysis demonstrated a diverse distribution of category 1 HIV-1-infected cells across the entire spectrum of mCD4+ T cells, with only minor under- or over-representation in specific cell clusters relative to HIV-uninfected cells; specifically, there was no evidence of enhanced clustering of virally infected category 1 cells in activated, HLA-DR+ or CD38+mCD4+ T cells (Extended Data Fig. 3a–c). A similar observation was made for category 4 cells (harbouring hypermutated proviruses), which closely imitated the phenotypic distribution pattern of category 0 and 1 cells (Fig. 2a,b). The phenotypic profiles of category 2 and 3 cells were markedly biased in comparison to those of category 0, 1 and 4 cells; cells from both categories 2 and 3 were disproportionally enriched within clusters exhibiting features of a more mature, effector memory phenotype, characterized by high levels of expression of CD45RO and low levels of expression of CCR7 and CD62L (Fig. 2a,b and Extended Data Fig. 3a–c). Together, these results indicate that cells encoding intact proviruses and/or being part of large proviral clones exhibit distinct phenotypic properties; by contrast, virally infected cells in categories 1 and 4, mostly consisting of lymphocytes harbouring defective proviruses, remained phenotypically largely indistinguishable from HIV-uninfected cells in this global phenotypic analysis.

For a formal statistical evaluation of the phenotypic profile of HIV-1-infected cells from PB, we first established an average read count for unspecific isotype control antibody binding, normalized to the total read count in each cell; this background was then used as a cutoff to calculate the proportion of cells expressing a given surface marker in each category of virally infected cells, relative to the HIV-uninfected cells (Supplementary Table 3). To account for differences in the total number of virally infected cells available from each participant, which could disproportionately bias results by participants with higher numbers of HIV-1-positive cells, we used a bootstrapping analysis approach so that data from each participant had an equal impact on final statistical outcomes. Several surface markers with significant enrichment in category 1 HIV-1-infected cells were noted, relative to HIV-uninfected cells (Fig. 2c). Although differences for many of these markers reached very definitive levels of statistical significance owing to relatively high numbers of cells in category 1, their biological significance was questionable, given that only very small proportions of category 1 cells were positive for such markers, and/or there were rather limited fold enrichments relative to uninfected cells. This was true for expression of, for example, CD276, GITR, CD57, interleukin-10 receptor (IL-10R), HLA-DR, LIGHT and OX40. Overall, inter-individually consistent surface marker expression differences between category 1 cells and HIV-uninfected cells were modest, suggesting that category 1 cells frequently show only slight phenotypic variations relative to their uninfected counterparts (Figs. 2c and 3a and Extended Data Fig. 4). Notably, weaker phenotypic differences were also observed for category 4 cells relative to HIV-uninfected cells (Fig. 3a and Extended Data Figs. 4 and 5a).

## Reservoir cells harbouring intact HIV-1

For PB cells harbouring proviruses with a high probability to be genome-intact (category 2 cells), we noted more pronounced statistical enrichments for a list of markers that are associated with functional inhibition of T cells (Figs. 2c and 3a and Extended Data Fig. 4). Such markers included the immune checkpoint receptors PD1, TIGIT, BTLA,

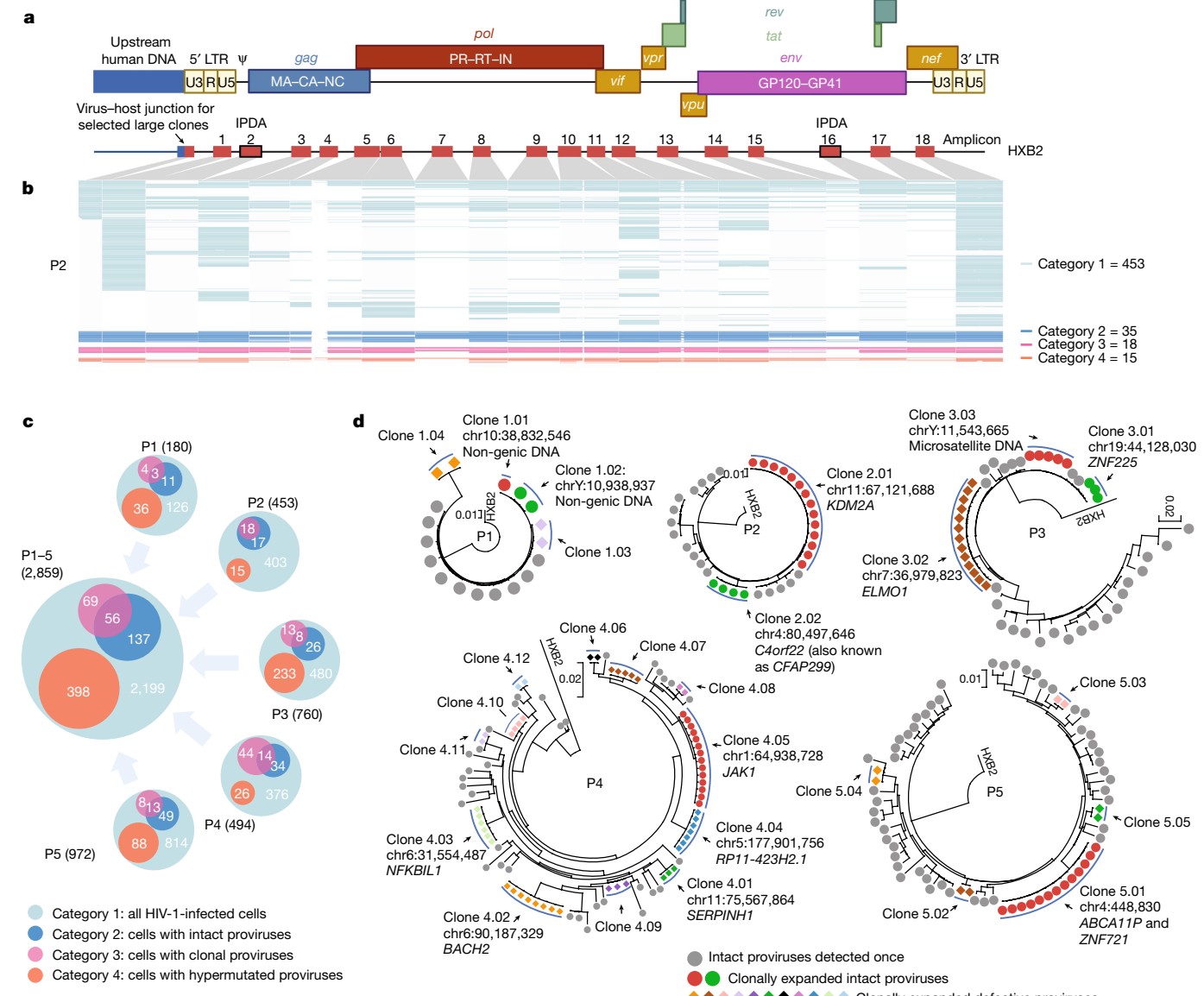

**Fig. 1 | Combined assessment of cellular phenotype and proviral sequence in HIV-1-infected cells (PheP-seq) from PB. a**, The genomic location of 18 different HIV-1 amplification products incorporated in the PheP-seq assay; these 18 amplicons cover approximately 4,000 bp of the HIV-1 genome; gaps between amplification products are indicated. Note that additional amplicons corresponding to predefined virus–host junctions of large proviral clones were also included. Amplicons 2 and 16 were previously described for the IPDA assay. LTR, long terminal repeat. Ψ, retroviral psi packaging element. **b**, A virogram summarizing individual HIV-1 DNA amplification products in single HIV-1-infected cells from study participant 2 (P2). Each row represents data from one infected cell; the numbers of sequences meeting criteria for categories 1, 2,

3 and 4 are shown. For longer amplification products (amplicons 5, 10 and 12), centrally located nucleotides could not be efficiently amplified using the Illumina NextSeq 2 × 150-bp sequencing read length used in this study. **c**, Venn diagrams summarizing the numbers of category 1–4 HIV-1-infected cells, shown separately for each study participant (P1–P5) and for all study participants combined. The numbers of sequences meeting criteria for different proviral categories are shown individually in each category. **d**, Circular maximum-likelihood phylogenetic trees of selected proviral sequences from each study participant, determined by PheP-seq. Clonal sequences are indicated; chromosomal integration sites are listed when available. Graphics in **a** were created using BioRender.com.

2B4 and KLRG1; only minor, nonsignificant changes were observed for CTLA4, TIM3 and LAG3. By increasing the threshold for cellular activation, these markers may limit proviral gene expression and reduce subsequent visibility and vulnerability of reservoir cells to host immune recognition mechanisms; pharmacological blockade of one checkpoint marker (PD1) was indeed shown to increase proviral gene expression in recent animal[26] and human clinical studies[27,28]. Notably, category 2 cells also showed elevated expression of several markers that act as ligands for inhibitory receptors expressed on immune effector cells and increase resistance of target cells to CD8[+] T or natural killer (NK) cell-mediated immune activity by conferring negative

immunoregulatory impulses (Figs. 2c and 3a); overexpression of such markers might defend HIV-1-infected cells against cellular immune responses and may translate into a longitudinal selection advantage. In particular, we observed upregulation of the herpes virus entry mediator (HVEM), which negatively regulates T and NK cells through binding to BLAT[29] and CD160 (refs. [30,31]) and has a known role for protecting virally infected cells against host immune effects, supported by the discovery of the HVEM paralogue UL144 encoded by human CMV[32] (Fig. 3a). In addition, we noted upregulation of the poliovirus receptor (PVR), which can inhibit cytotoxic effects of T and NK cells by interactions with its high-affinity ligands TIGIT[33] and KIR2DL5 (ref. [34]), respectively.

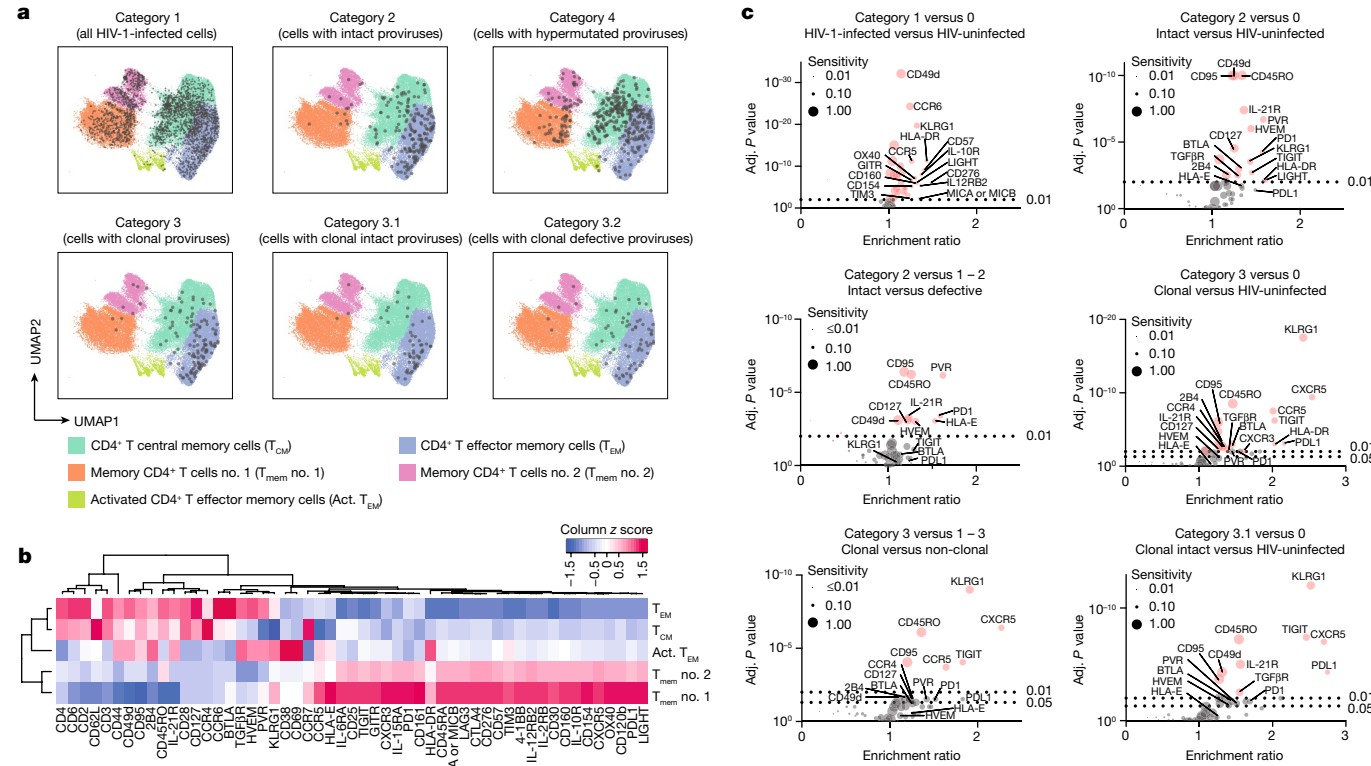

**Fig. 2 | Phenotypic profile of patient-derived HIV-1-infected cells circulating in PB. a**, Global visualization of phenotypic properties of HIV-1-infected CD4⁺ T cells from five study participants using two-dimensional UMAP plots; five computationally defined spherical clusters reflecting phenotypically distinct mCD4⁺ T cell subsets are shown. One plot is shown for each cell category. **b**, A heatmap summarizing the normalized phenotypic profile of cells in each spherical cluster, based on 53 surface markers included in this study. **c**, Volcano plots showing enrichment ratios of marker-positive

cells and corresponding FDR-adjusted (adj.) *P* values for all 53 surface markers included in this study; selected markers are labelled individually. Marker sensitivities, calculated as the proportions of marker-positive cells in the indicated categories of cells, are indicated by dot sizes. Comparisons between indicated categories of cells are depicted; bootstrapped data from all five participants are shown. Significance was tested using a two-sided chi-squared test; FDR-adjusted *P* values are shown.

Enrichment within category 2 cells was also observed for PDL1 (the physiological ligand for PD1) and for HLA-E, a non-classical MHC class I molecule that acts as the high-affinity ligand for the inhibitory receptor NKG2A expressed on NK and T cells[35]; however, these markers were expressed on more limited numbers of cells and statistical scores were weaker (false discovery rate (FDR)-adjusted *P* = 0.02 for HLA-E and *P* = 0.04 for PDL1). Category 2 cells harbouring intact HIV-1 DNA were further enriched for a higher level of surface expression of CD49d, CD45RO and CD95, all of which can be associated with T cell activation and differentiation towards an effector memory profile. Moreover, they exhibited elevated expression of the receptor for transforming growth factor β (TGFβ), which can promote proviral latency[36], of the receptor for IL-21, and of CD127, the receptor for the homeostatic cytokine IL-7. In a subsequent analysis, we noted little phenotypic variation between cells harbouring clonal genome-intact proviruses and intact proviruses detected only once, with the notable exception of KLRG1, which was more strongly expressed on the former (Extended Data Fig. 4a). Moreover, we compared the phenotypic profiles of category 2 cells with those of cells encoding for defective proviruses (category 1 cells after excluding category 2 cells; Figs. 2c and 3a); these analyses involved lower numbers of cells and reached lower levels of statistical significance, but also identified elevated surface expression of markers associated with resistance to immune-mediated killing (PVR, HLA-E and HVEM) as key distinguishing features for cells enriched for harbouring genome-intact HIV-1 DNA. Markers associated with functional inhibition of T cells, in particular PD1, were also upregulated in category 2 cells relative to cells harbouring defective proviruses (Figs. 2c and 3a).

## Phenotype of clonal reservoir cells

In a dedicated analysis of cells harbouring clonally expanded proviruses (category 3), we noted strong enrichment for expression of KLRG1, which, in addition to its role as an inhibitory checkpoint marker binding to E-cadherin[37], has been associated with terminal differentiation and cellular senescence[38]; enrichment of KLRG1 in category 3 cells probably reflects a history of strong clonal proliferative turnover in this specific cell population (Figs. 2c and 3). PDL1 emerged as a negative immunoregulatory marker with a more notable enrichment on category 3 cells, suggesting a role of this marker for protecting clonal HIV-1-infected target cells against T cell immune attacks. Among markers associated with functional inhibition of T cells, TIGIT exhibited the most notable upregulation in category 3 cells; expression of PD1 and BTLA was also increased on category 3 cells. For a more detailed analysis of individual clones of HIV-1 reservoir cells, we investigated the phenotype of clonal HIV-1-infected cells sharing a common chromosomal integration site. In a global analysis, cells belonging to the same clone tended to cluster near one another on a UMAP plot, suggesting a rather homogenous phenotypic behaviour of HIV-1-infected cells derived from the same clone (Fig. 4a); however, some clones showed a higher level of phenotypic variability. Moreover, we noted that the upregulation of specific immunoregulatory markers on category 3 cells, including PDL1, HVEM and PVR, was not selectively driven by specific cell clones but occurred relatively consistently across most analysed clonal reservoir cell populations (Fig. 4b); nevertheless, some inter-clonal phenotypic diversity was noted and deserves additional

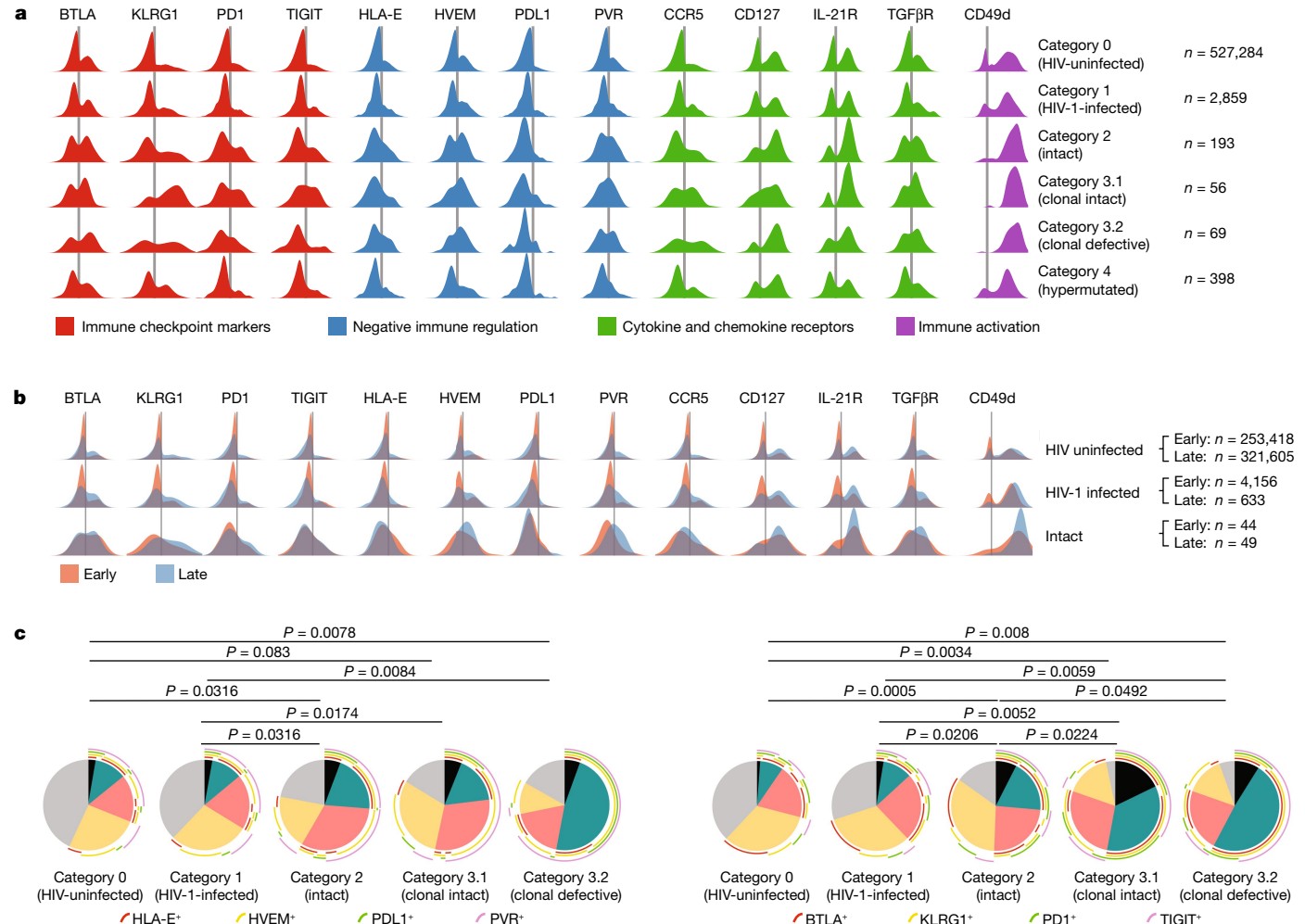

**Fig. 3 | Differentially expressed surface markers on patient-derived HIV-1-infected CD4⁺ T cells circulating in PB. a**, Density plots indicating the surface expression of selected phenotypic markers on indicated categories of cells in a cross-sectional analysis from all five participants after approximately 10 years of ART. **b**, Density plots reflecting surface expression of selected phenotypic markers in longitudinal samples collected at approximately year 1 ('early') and year 10 ('late') of ART in study participants P1 and P2. **c**, 'Simplified Presentation of Incredibly Complex Evaluations' (SPICE) diagrams reflecting the proportions of HIV-1-infected cells expressing ensemble phenotypic markers in a cross-sectional analysis of samples collected from five study participants after 10 years of ART. The pie charts indicate the relative proportions of cells expressing 0, 1, 2, 3 or 4 of the indicated markers; individual markers are shown as overlaying arches. One separate diagram is shown for each category of cells; the data in the left panel reflect phenotypic markers associated with resistance to immune-mediated killing, and the data in the right panel show immune checkpoint markers. Significance was tested using an FDR-adjusted permutation test.

investigation in future studies. Together, these findings suggest that clonal HIV-1-infected cells, which may be under more profound immune selection pressure due to a higher propensity for viral reactivation during cellular proliferation, may exhibit a specific surface phenotype that is likely to increase a cell's ability to resist host immune selection forces.

## Combinatorial and longitudinal analysis

We subsequently considered a combinatorial analysis of surface markers that may be jointly upregulated on HIV-1-infected cells from the PB, applying an algorithm previously developed for analysing multi-variate flow cytometry datasets[39]. These additional studies demonstrated that after prolonged periods of viral suppression, proportions of cells expressing none of the markers associated with protection from killing by T or NK cells (HVEM, PVR, HLA-E and PDL1) were significantly higher in category 0 and category 1 cells, whereas the fractions of cells simultaneously expressing two or more of these markers were significantly increased among category 2 and 3 cells (Fig. 3c).

A similar observation was made for immune checkpoint molecules (PD1, KLRG1, BTLA and TIGIT); by contrast, a combinatorial analysis failed to demonstrate a pronounced enrichment of category 2 or 3 cells with combinations of markers associated with immune activation (CD38, CD49d, CD95, CD69 and HLA-DR; Extended Data Fig. 5b). These findings were corroborated by experiments evaluating the frequencies of intact proviruses in sorted mCD4⁺ T cells expressing the described marker combinations, although some variations in phenotypic patterns were noted among different study participants (Extended Data Figs. 6 and 7). Collectively, our results indicate that category 2 and 3 reservoir cells are characterized by ensemble phenotypic signatures of immune checkpoint molecules that probably reduce a cell's propensity to reactivate proviral gene transcription, and of marker combinations that can protect infected target cells against killing by T and NK cells. Such a phenotypic profile seems evolutionarily advantageous by minimizing exposure to and killing by host immune effector cells and is consistent with functional studies suggesting increased resistance of HIV-1 reservoir cells to T cell-mediated killing[40,41]. Functional

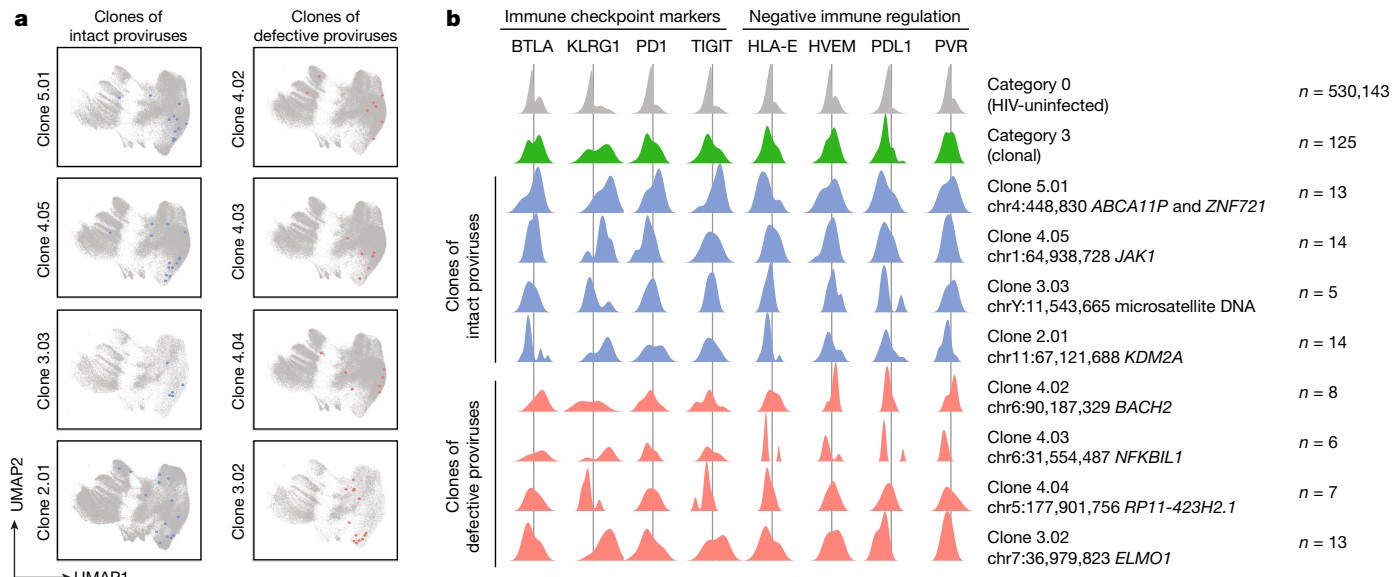

**Fig. 4 | Phenotypic profile of individual HIV-1-infected CD4+ T cell clones circulating in PB. a**, Two-dimensional UMAP diagrams indicating the global phenotypic profile of selected HIV-1-infected CD4+ T cell clones relative to HIV-uninfected mCD4 T cells. **b**, Density plots showing the surface expression of selected phenotypic markers on individual HIV-1-infected CD4+ T cell clones, identified by proviral chromosomal integration sites. The data were collected in a cross-sectional analysis from study participants after approximately 10 years of ART. Results from all clonal HIV-1-infected cells (category 3) and HIV-uninfected cells (category 0) are shown as references.

experiments conducted with cells from our study participants also suggested enhanced resistance of reservoir cells encoding for intact HIV-1 DNA to HIV-1-specific CD8+ T cell-mediated killing (Extended Data Fig. 8).

To longitudinally evaluate the stability of the reported phenotypic profile of HIV-1 reservoir cells, we carried out PheP-seq assays on PB samples collected early (1–2 years) after ART initiation in study participants 1 and 2, from whom such samples were available for analysis (Extended Data Fig. 2). In total, $n = 257,574$ individual mCD4+ T cells were analysed from these early time points, of which $n = 4,156$ and $n = 44$ were classified as category 1 and 2 cells, respectively. Notably, category 2 reservoir cells already exhibited an increased level of expression of immune checkpoint markers and ligands for inhibitory receptors of T or NK cells at early time points after ART initiation, although such changes tended to be less pronounced relative to later time points of ART for some markers, specifically for PVR and HVEM (Fig. 3b and Extended Data Fig. 5c). Moreover, there was a trend for elevated expression for the receptors of IL-21 and TGFβ after more extended durations of ART. Together, these observations suggest that selection of reservoir cells starts early after ART initiation, possibly implying that only a small subset of category 2 cells with specific phenotypic properties are pre-destined to enter the long-lasting reservoir cell pool. As after longer durations of ART, category 1 reservoir cells from the early time point exhibited only limited phenotypic variations relative to uninfected cells from the same time point. Few cells analysed at early stages of ART meet our criteria for category 3 reservoir cells, precluding a specific comparison of such cells between early and late ART.

## HIV-1 reservoir cells in lymph nodes

As most HIV-1 reservoir cells are located in lymphoid tissues[42], we subsequently conducted a phenotypic analysis of virally infected cells from inguinal lymph nodes (LNs) collected by surgical excision from three people living with HIV-1 who remained on continuous suppressive ART for approximately 10–15 years (Extended Data Fig. 9a,b).

In total, we analysed the phenotype of $n = 396,628$ single mCD4+ T cells from these samples, of which $n = 3,888$, $n = 111$ and $n = 39$ met criteria for category 1, 2 and 4 cells, respectively; the small number of category 4 cells may be related to primer mismatches to hypermutated proviral sequences. Category 3 cells were not individually analysed in LN samples owing to the limited number of cells meeting our category 3 criteria (Extended Data Fig. 9c). A global analysis of the phenotypic landscape of HIV-1 reservoir cells from LNs (Fig. 5a,b) indicated that viral reservoir cells exhibited focal enrichments in two different cell clusters, one characterized by high-level surface expression of CXCR5, indicative of T follicular helper ($T_{FH}$) cell polarization[43], and a second one characterized by elevated expression of CD127 and CD69, suggestive of a CD4+ tissue resident memory T ($T_{RM}$) cell profile in LNs[44,45]. Whereas $T_{FH}$ cells accounted for most cells in category 1, consistent with prior results[46], category 2 cells were more equally balanced between the $T_{FH}$ cells and $T_{RM}$ cells. Subsequently, we conducted a detailed evaluation of differentially expressed surface markers between the different categories of HIV-1-infected cells, again using a bootstrapping analysis approach ensuring equal contribution of each study participant to statistical outcomes (Fig. 5c, Extended Data Fig. 10a,b and Supplementary Table 3). These evaluations demonstrated upregulation of some immune checkpoint markers on category 2 cells in LNs; however, statistical scores and enrichment ratios were weaker relative to category 2 reservoir cells from PB. Similarly, fewer markers conferring resistance to killing by T or NK cells were upregulated on category 2 cells in LNs relative to category 2 cells from PB, although both PVR and HLA-E met statistical criteria for upregulation on category 2 cells in LNs. These differential phenotypic profiles between category 2 cells from LNs versus blood may possibly be related to the predominant presence of non-cytolytic CD8+ T cells in LNs[47,48], probably resulting in a more limited selection advantage for viral reservoir cells with phenotypic signs of increased resistance to immune-mediated killing in this tissue compartment. Markers that strongly distinguished category 2 reservoir cells from uninfected cells in LNs included CD44, the IL-21 receptor (CD360), the IL-7 receptor (CD127) and, to a lesser extent,

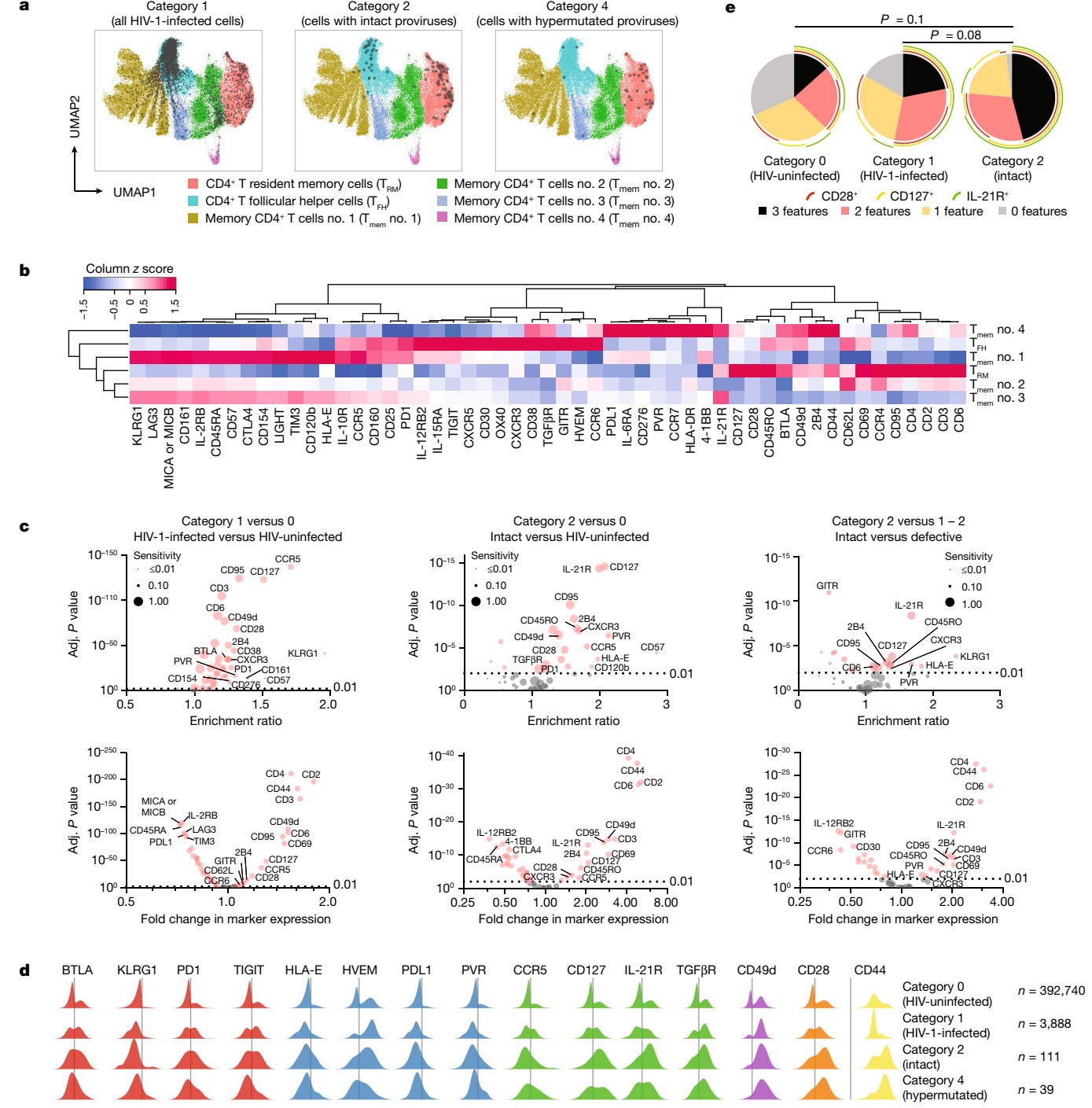

**Fig. 5 | Phenotypic analysis of HIV-1 reservoir cells from inguinal LNs.**
**a**, Global phenotypic analysis of HIV-1 reservoir cells from LNs by two-dimensional UMAP plots. Six computationally defined spherical clusters are indicated; one plot is shown each for category 1, category 2 and category 4 cells. **b**, A heatmap summarizing the phenotypic profile of cells in each spherical cluster, based on 53 surface markers included in this study. **c**, Volcano plots showing enrichment ratios of marker-positive cells (upper panel) or fold changes in marker expression intensity (lower panel) and corresponding FDR-adjusted *P* values for all 53 surface markers included in this study; selected markers are labelled individually. Marker sensitivities, calculated as the proportions of marker-positive cells in the indicated categories of cells, are

indicated by dot sizes. Comparisons between indicated categories of cells are depicted; bootstrapped data from all three donors of LN biopsies are shown. Significance was tested using a chi-squared test or two-sided *t*-test, FDR-adjusted *P* values are shown. **d**, Density plots indicating the expression of selected phenotypic markers on indicated categories of cells from all three LN donors. **e**, SPICE diagrams reflecting the proportions of HIV-1-infected LN cells expressing the ensemble phenotypic cell survival markers CD127, CD28 and IL-21R. The pie charts indicate the relative proportions of cells expressing none, one, two or three of the indicated markers; individual markers are shown as overlaying arches. One separate diagram is shown for each indicated population of cells. Significance was tested using an FDR-adjusted permutation test.

CD28 (Fig. 5c,d); these markers were also upregulated on category 2 reservoir cells in LNs relative to LN cells harbouring defective proviruses (Fig. 5c) and to category 1 and 2 reservoir cells from PB (Extended Data Fig. 10B). All of these markers are functionally involved in mediating cell survival signals; in particular, CD44 can increase survival and apoptosis resistance of CD4[+] T cells through engagement of the phosphoinositide 3-kinase–AKT signalling pathway[49]. Moreover, T cell survival signals related to activation of the AKT signalling pathway also occur downstream of the IL-21 receptor[50,51]. Likewise, CD28 has a prominent role for supporting CD4[+] T cell survival; this is probably related to a CD28-mediated upregulation of cell-intrinsic anti-apoptosis markers from the BCL-2 family[52]. Phenotypic signatures of increased cell survival were also detected on category 2 cells in LNs when a combinatorial analysis was applied (Fig. 5e). Notably, we also observed a trend for elevated expression of several members of the TNF receptor superfamily on category 2 cells in LNs, specifically of CD30, GITR and OX40, which have also been implicated in mediating cell survival signals[53] (Extended Data Fig. 10a). Phenotypic differences between category 2 and category 1 or 4 reservoir cells were not as definitive in LNs compared to PB cells, a finding that requires more attention in future studies. Together, our data demonstrate that category 2 reservoir cells from LNs predominantly show phenotypic features associated with apoptosis resistance and survival; upregulation of these markers on category 2 cells in LNs may promote their long-term persistence and can be viewed as a sign of immune adaptation of reservoir cells to the LN immune microenvironment.

## Discussion

Our experiments, involving single-cell proteogenomic profiling data from a total of 1,184,345 individual patient-derived mCD4[+] T cells, indicate that following years of continuous ART, HIV-1-infected cells show several distinct phenotypic features, specifically when harbouring intact proviruses; however, they do not support the presence of a single unifying phenotypic marker that can effectively distinguish virally infected cells from uninfected cells. Given that a higher expression level of certain markers was more definitively observed on category 2 (in LNs and PB) and on category 3 cells (in PB), we propose that these phenotypic changes are not due to an intrinsic association with HIV-1 infection, or a higher susceptibility of cells expressing these markers to HIV-1 infection. Instead, they probably represent a consequence of immune selection mechanisms that promote preferential persistence of reservoir cells with a phenotypic profile offering optimal adaptation to their specific immunological microenvironment. We emphasize that the enrichment for the described phenotypic markers, specifically when analysed as isolated parameters, must be considered as moderate, although there were stronger enrichments for specific marker combinations, suggesting that phenotypic marker ensembles allow for a more effective and, possibly, a more customized adaptation to host immune activity. Together, our findings strongly support the hypothesis that viral reservoir cells are subject to active host immune selection pressure, and that footprints of such immune selection are particularly visible in cells harbouring intact HIV-1 proviruses. Given their likely dependence on host immune selection activity, we suggest that phenotypic alterations of viral reservoir cells described here are unlikely to act as universal biomarkers of viral reservoir cells in all ART-treated people living with HIV-1; instead, the presence or absence of such phenotypic selection footprints may vary among different people and might be critically influenced by the intensity of host immune activities; evaluations of the longitudinal coevolution of host immune responses and phenotypic profiles of viral reservoir cells may represent an attractive future research perspective for understanding viral reservoir dynamics. Although we do not claim that the phenotypic markers identified here may represent direct targets for immunotherapeutic interventions in HIV-1 cure research, we suggest that clinical strategies designed to intensify and accelerate immune selection of HIV-1 reservoir cells may be of benefit for reducing HIV-1 long-term persistence and inducing a drug-free remission of HIV-1 infection.

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

## Methods

### Study participants

Individuals infected with HIV-1 were recruited as study participants at the Massachusetts General Hospital in Boston, MA, and at the NIH Clinical Center in Bethesda, MD. PB mononuclear cell (PBMC) samples were obtained according to protocols approved by the respective Institutional Review Boards. All study participants gave written informed consent for blood collection or for LN biopsies. Clinical characteristics of study participants are summarized in Extended Data Figs. 2 and 9. Study participants were preselected on the basis of high frequencies of HIV-1-infected CD4[+] T cells analysed in previous studies.

### LN biopsies

Inguinal LNs were excised surgically with informed consent of study participants, according to protocols approved by the Massachusetts General Hospital Institutional Review Board. LN tissue was dissected and mechanically disaggregated through a 70-μm nylon cell strainer in RPMI medium supplemented with 10% fetal bovine serum.

### Isolation of memory CD4[+] T cells

PBMCs and LN mononuclear cells (LNMCs) were isolated using Ficoll–Paque density centrifugation. PBMCs and LNMCs were viably frozen in 90–95% fetal bovine serum and 5–10% dimethylsulfoxide. For analysis, cells were thawed and subjected to negative immunomagnetic isolation of memory CD4[+] T cells, using a commercial product (Stemcell EasySep Human Memory CD4[+] T cell Enrichment Kit, no. 18000) per the manufacturer's protocol.

### Surface labelling with monoclonal antibodies

Monoclonal antibodies tagged with distinct oligonucleotides were custom manufactured and supplied as lyophilized single-reaction vials by a commercial vendor (BioLegend). Antibodies to the following surface markers were used: PDL1 (clone 29E.2A3), CD276 (clone DCN.70), HVEM (clone 122), CD155 (clone SKII.4), CD154 (also known as CD40L) (clone 24-31), CCR4 (clone L291H4), PD1 (A17188B), TIGIT (clone A15153G), CD44 (clone BJ18), CXCR3 (clone G025H7), CCR5 (clone HEK/1/85a), CCR6 (clone G034E3), CXCR5 (clone J252D4), CCR7 (clone G043H7), KLRB1 (also known as CD161) (clone HP-3G10), CTLA4 (clone BNI3), LAG3 (clone 11C3C65), KLRG1 (clone 14C2A07), CD95 (clone, DX2), OX40 (also known as CD134) (clone Ber-ACT35), CD57 (clone HNK-1), TIM3 (clone F38-2E2), BTLA (also known as CD272); (clone MIH26), CD244 (also known as 2B4) (clone 2-69), IL-2R (clone TU27), CD137 (also known as 4-1BB) (clone 4B4-1), GITR (also known as CD357) (clone 108-17), CD28 (clone CD28.2), CD127 (clone A019D5), IL-6R (clone UV4), HLA-E (clone 3D12), MICA or MICB (clone 6D4), IL-15R (also known as CD215) (clone JM7A4), IL-21R (clone 2G1-K12), TNFR2 (clone 3G7A02), CD160 (clone BY55), LIGHT (also known as CD258) (clone T5-39), IL-10R (also known as CD210) (clone 3F9), TGFβR (clone W17055E), IL-12R (clone S16020B), CD6 (clone BL-CD6), CD49d (clone 9F10), CD25 (clone BC96), CD30 (clone BY88), CD69 (clone FN50), CD45RA (clone HI100), CD38 (clone HIT2), HLA-DR (clone L243), CD4 (clone RPA-T4), CD2 (clone TS1/8), CD3 (clone UCHT1), CD62L (clone DREG-56), CD45RO (clone UCHL1). Two mouse IgG control antibodies (BioLegend, no. 400299, no. 400383), conjugated with distinct TotalSeq-D oligonucleotides were included as the isotype controls. The lyophilized antibody cocktails containing the above-mentioned antibodies were reconstituted with 60-μl of cell staining buffer (BioLegend, no. 420201). Cells were blocked with Human TruStain FcX (BioLegend, no. 422301) for 15 min on ice and then incubated with 50 μl of reconstituted antibody cocktail for 30 min on ice. After three washes with pre-chilled cell staining buffer, cells were filtered with a 40-μm cell Flowmi strainer (Fisher Scientific, no. 14-100-150), counted with an automated cell counter, and then loaded into a microfluidic cartridge for single-cell multiplex PCR assays.

### Single-cell multiplex PCR

Single-cell amplification of defined genomic DNA segments was carried out using the Tapestri platform (MissionBio) according to the manufacturer's protocol[18]. Viable single cells were encapsulated into droplets with a lysis buffer (containing protease and a mild detergent) and incubated for 1 h at 50 °C, followed by 10 min at 80 °C for heat inactivation of enzymes. Droplets containing single-cell barcoding beads (tagged with oligonucleotides carrying the cellular barcodes and custom-designed primers) were fused with encapsulated cell lysates. A panel of primers designed to amplify $n = 18$ different genomic regions in HIV-1 and $n = 27$ specific HIV–host DNA junctions of previously defined large clonal HIV-1-infected T cell populations in our study participants were used (Fig. 1b and Supplementary Table 1) in addition to two primer sets amplifying control genomic regions in the *RPP30* gene on chromosome 10. The droplets were placed under ultraviolet light to cleave PCR primers containing unique cell barcodes from beads. To amplify the selected genomic DNA segments and the antibody oligonucleotide tags, droplets were subjected to PCR for 24 cycles with temperature gradients recommended by the manufacturer.

### Sequencing library construction

Amplification products were pooled, mixed with AMPure XP beads (Beckman Coulter, no. A63882) at a ratio of 0.7 and placed in a magnetic field for separating the DNA and the protein tag libraries. The DNA library bound to AMPure beads was washed with 80% ethanol, and the supernatant containing the protein tag library was aspirated and incubated with a biotinylated oligonucleotide complementary to the 5′ end of the antibody tags, followed by magnetic isolation using streptavidin beads. For library amplification, PCRs were carried out with Illumina index primers P5 and P7 on purified DNA and protein libraries, respectively, according to the manufacturer's protocol; 12 cycles were carried out for the DNA library, and 18 cycles were run for the protein tag library.

### Next-generation sequencing

The DNA and protein tag libraries were run on a High Sensitivity D1000 ScreenTape instrument (Agilent Technologies, 5067-5584) with the Agilent 4200 TapeStation System to evaluate DNA quality. Libraries were quantified by a fluorometer (Qubit 4.0, Invitrogen) and sequenced on Illumina next-generation sequencing platforms with a 20% spike-in of PhiX control DNA (Illumina, no. FC-110-3002). DNA and protein tag libraries were sequenced separately on a NextSeq 500 instrument (Illumina), using the NextSeq 500/550 High Output flow cell v2.5 (Illumina, no. 20022408) and the NextSeq 500/550 High Output Kit v2.5 (300 cycles; Illumina, no. 20024908) in 2 × 150-bp paired-end runs.

### Bioinformatic analysis

The Tapestri pipeline (MissionBio, v2.0.1) with minor modifications was used to process the sequencing data. Briefly, for DNA library data, cutadapt (v2.5)[54] was used to trim 5′ and 3′ adaptor sequences, and extract 18-bp cell barcode sequences from read 1. Cell barcodes that aligned to a unique barcode on a whitelist within a Hamming distance of 2 were used for downstream analysis. Using bwa (v0.7.12)[55], sequences were aligned to custom reference genomes built from the human genome (GRCh38) and patient-specific autologous HIV-1 sequences identified in prior studies. Single-cell alignments were filtered according to criteria implemented in the Tapestri pipeline, and indexed using samtools (v1.9)[56]. Candidate HIV-1-infected cells were determined by the CellFinder algorithm built in the Tapestri pipeline. Bcftools (v1.9)[57] was used to call variants and generate consensus sequences. To reduce spurious alignments, viral sequencing reads were considered valid only if they covered at least 80% of the length of the reference sequence for each given amplicon; host sequencing reads had to cover at least 50% of the length of respective amplicon. For antibody library data,

cell barcodes were similarly extracted using cutadapt. For reads with valid cell barcodes, 15-bp antibody barcodes were extracted from read 2. Antibody barcode sequences within a Hamming distance of 1 from known antibody barcodes were accepted. Candidate cells appearing in both libraries were processed for downstream analysis. Read counts for each antibody were normalized to the total read count in each cell using centred log-ratio transformation[58]. Cutoffs for a given phenotypic marker to be considered positive were defined by marker-specific read counts higher than 1 mean absolute deviation of the normalized median read count corresponding to unspecific IgG control antibodies. To generate a more homogenous cell population for analysis, centred log-ratio values of all candidate cells that were CD3−CD4− (non-CD4+ T cells) and CCR7+CD45RA+ (contaminating naive CD4+ T cells) were excluded.

## Dimension reduction and clustering

UMAP embeddings in two dimensions of the centred log-ratio values was carried out through the Monocle 3 (ref. [59]) and uwot[60] packages with the number of principal components set to 15 for all cells, and to 10 for HIV-1-infected cells alone; numbers of neighbours to use during $k$-nearest neighbours graph construction were set to 9 and 5, respectively. All other settings were kept to the default values. The cells were clustered using the Leiden community detection algorithm through Monocle 3, with the $k$-near neighbours set to 500.

## Phylogenetic analysis of HIV-1

HIV-1 sequencing reads corresponding to each of the 18 HIV-1 amplification products and of additional amplification products corresponding to predefined virus–host junctions were aligned to the reference HIV-1 genome HXB2, to autologous intact HIV-1 sequences and to the human reference genome GRCh38. The presence or absence of hypermutations associated with APOBEC3G or APOBEC3F was determined using the Los Alamos National Laboratory HIV Sequence Database Hypermut 2.0 program[61]. Sequence alignments were carried out using MUSCLE[62]. Phylogenetic distances between sequences were examined using maximum-likelihood trees in MEGA (https://www.megasoftware. net) and MAFFT (https://mafft.cbrc.jp/alignment/software), and visualized using highlighter plots (https://www.lanl.gov). Proviruses were classified in 4 different categories according to the following criteria—category 1 (any provirus): at least 20 total valid viral sequencing reads with at least 4 reads in at least 2 different HIV-1 amplicons each; category 2 (enriched for intact proviruses): a total of at least 20 viral sequencing reads with at least 4 sequencing reads from amplicons 2 and 16 (corresponding to IPDA amplicons) each or at least 4 reads from at least 15 amplification products each or at least 4 reads from amplification products spanning known virus–host junctions of intact proviruses (all viral sequencing reads in category 2 had to be without evidence of statistically significant hypermutation and could not correspond to virus–host junctions from known defective proviruses); category 3 (clonal proviruses): proviruses with identical proviral integration sites (based on at least 4 viral sequencing reads from amplification products of known virus–host junctions) or completely identical proviral sequences; category 4 (hypermutated proviruses): proviral sequences from category 1 that exhibited statistically significant sequence hypermutations (FDR-adjusted $P < 0.05$). Cells with HIV-1 sequencing reads that did not meet any of the above-mentioned criteria were excluded from the analysis. Category 0 cells were defined by complete absence of sequencing reads corresponding to HIV-1.

## Cell sorting and flow cytometry

Memory CD4+ T cells isolated from PBMCs by negative immunomagnetic enrichment were incubated with defined fluorophore-labelled surface antibodies: CD3–PerCP–Cy5.5 (BioLegend, clone UCHT1, catalogue no. 300430), CD4–BUV395 (BD, clone RPA-T4, catalogue no. 564724), PD1–FITC (BioLegend, clone A17188B, catalogue no. 621612), TIGIT–BV421 (BioLegend, clone A15153G, catalogue no. 372710), PVR–PE

(eBioscience, clone 2H7CD155, catalogue no. 12-1550-41), HVEM–APC (BioLegend, clone 122, catalogue no. 318808), BTLA–FITC (BioLegend, clone MIH26, catalogue no. 344524), KLRG1–APC (BioLegend, clone 14C2A07, catalogue no. 368606), HLA-E–BV421 (BioLegend, clone 3D12, catalogue no. 342612), PDL1–PE (BioLegend, clone MIH2, catalogue no. 393608). After 25 min of incubation, cells were washed, and indicated cell populations were sorted in a specifically designated biosafety cabinet (Baker Hood), using a FACSAria cell sorter (BD Biosciences) at 70 pounds per square inch. Cell sorting was carried out by the Ragon Institute Imaging Core Facility at Massachusetts General Hospital. Data were analysed using FlowJo software (Tree Star). Sorted cells were subjected to proviral sequencing analysis using the full-length individual proviral sequencing assay (FLIP-seq).

## Full-length HIV-1 sequencing assay

A previously described protocol was used[11,23]. In brief, genomic DNA was extracted from sorted cell populations using a QIAGEN DNeasy Blood & Tissue kit. DNA diluted to single-genome levels based on Poisson distribution statistics was subjected to single-genome amplification using Invitrogen Platinum Taq and nested primers spanning near-full-length HIV-1. PCR products were visualized by agarose gel electrophoresis; amplification products were subjected to single-genome sequencing on the Illumina platform. Resulting short reads were de novo assembled and aligned to HXB2 to identify large deleterious deletions, out-of-frame indels, premature/lethal stop codons, internal inversions or packaging signal deletions, using an automated in-house pipeline (https://github.com/BWH-Lichterfeld-Lab/Intactness-Pipeline). The presence or absence of hypermutations associated with APOBEC3G or APOBEC3F was determined using the Los Alamos HIV Sequence Database Hypermut 2.0 program. Viral sequences that lacked all lethal defects listed above were classified as genome-intact. Sequence alignments were carried out using MUSCLE. Phylogenetic distances between sequences were examined using Clustal X-generated neighbour-joining algorithms. Proviral species that were completely sequence-identical were considered as clonal.

## Ex-vivo HIV-1 reservoir cell killing assay

Memory CD4+ T cells isolated from PBMCs by negative immunomagnetic enrichment were incubated in R10 medium with or without epitopic peptides for 1 h and washed thoroughly. Afterwards, cells were co-incubated with previously isolated HIV-1-specific CD8+ T cell clones at selected effector-to-target (E/T) ratios for 16 h. After washing, cells were stained with blue viability dye (Invitrogen, catalogue no. L34962), CD3–FITC (BioLegend, clone UCHT1, catalogue no. 300406) and CD4–BV711 (BioLegend, clone RPA-T4, catalogue no. 300558) antibodies, followed by sorting of viable CD3+CD4+ events. Sorted cells were subjected to assessments of intact and total HIV-1 proviruses using the IPDA assay.

## IPDA

The IPDA uses digital droplet PCR to quantify proviruses lacking overt fatal defects, especially large deletions and hypermutations, and was carried out as previously described[22].

## Statistical analysis

The data are presented as pie charts, Venn diagrams, volcano plots, UMAP plots and heatmaps. Enrichment ratios between two cell populations for each marker were calculated as the ratio of the proportion of marker-positive cells in the first population divided by the proportion of marker-positive cells in the second population. Sensitivity values were calculated by dividing the number of marker-positive cells in each category of cells by the total number of cells in the respective category. A bootstrapped dataset was constructed by resampling equal numbers of cells from each cell category from each participant, while keeping the total number of cells from each cell category equal between the

bootstrapped and raw datasets. Differences were tested for statistical significance using Fisher's exact test, chi-squared test, $t$-test or Mann–Whitney $U$-test, as appropriate. $P$ values of <0.05 were considered significant; FDR correction was carried out using the Benjamini–Hochberg method[63] or the Bonferroni method. Analyses were carried out using Prism (GraphPad Software, Inc.), SPICE[39], R (R Foundation for Statistical Computing[64]) and Python (Python Software Foundation). Figures were generated using Adobe Illustrator.

## Reporting summary

Further information on research design is available in the Nature Portfolio Reporting Summary linked to this article.

## Data availability

Data supporting conclusions in this Article are provided in Supplementary Tables 1–3. Owing to study participant confidentiality concerns, viral sequencing data cannot be publicly released, but will be made available to investigators upon reasonable request and after signing a data-sharing agreement.

## Code availability

No new computational codes are reported in this Article.

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

**Acknowledgements** M.L. is supported by NIH grants AI117841, AI120008, AI130005, DK120387, AI152979, AI155233, AI135940 and AI169768, by the American Foundation for AIDS Research (amfAR, no. 110181-69-RGCV) and by the Campbell Foundation. X.G.Y. is supported by NIH grants AI155171, AI116228, AI078799, HL134539 and DA047034, amfAR ARCHE grant no. 110393-72-RPRL and the Bill and Melinda Gates Foundation (INV-002703). M.L. and X.G.Y. are members of the DARE, ERASE, PAVE and BEAT-HIV Martin Delaney Collaboratories (UM1 AI164560, AI164562, AI164566 and AI164570). Collection of LN samples was supported by a donation from M. and L. Schwartz.

**Author contributions** Design and performance of PheP-seq assay: W.S., M.R.O., K.B.E., C.G., X.G.Y., M.L. Cell sorting: W.S., C.A.H., N.B. HIV-1 reservoir cell killing assay: W.S., C.A.H. HIV-1 sequencing assays: W.S., X.L., B.B. Biological specimen contribution: E.S.R., T.-W.C., B.D.W. Bioinformatics analysis: C.G. Data analysis, interpretation, presentation: C.G., W.S., M.R.O., C.A.H., X.G.Y., M.L. Study concept and supervision: X.G.Y., M.L. W.S. and C.G. contributed equally to this work and shared the first authorship; C.A.H. and M.R.O. contributed equally to this work and shared the second authorship.

**Competing interests** The authors declare no competing interests.

**Additional information**
**Correspondence and requests for materials** should be addressed to Mathias Lichterfeld.

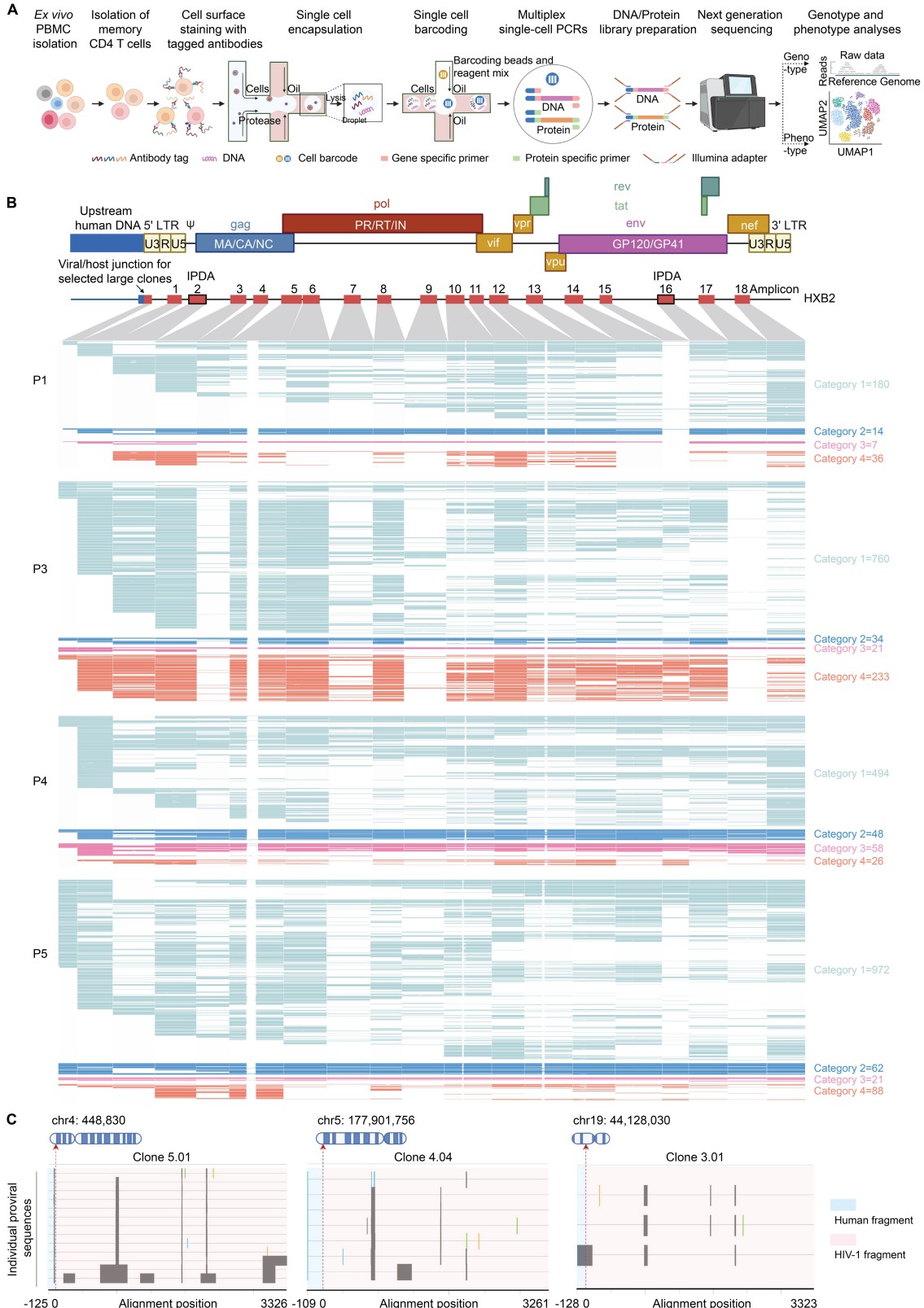

**Extended Data Fig. 1 | Single-cell proteogenomic profiling of HIV-1 reservoir cells from peripheral blood.** (A): Schematic representation of the experimental workflow for the PheP-seq assay. (B): Virograms summarizing individual HIV-1 DNA amplification products in single HIV-1-infected cells from study participants 1, 3, 4 and 5. Each row represents data from one infected cell; numbers of sequences meeting criteria for categories 1, 2, 3 and 4 are shown. For longer amplification products (amplicons 5, 10, 12), centrally located nucleotides could not be efficiently amplified using the Illumina

NextSeq 2 x 150-bp sequencing read length used in this study. (C): Highlighter plots reflecting representative clonal proviral sequences identified by PheP-seq. Sequencing data from all 18 amplification products from each single cell were fused. Sequencing reads corresponding to defined virus-host junctions and corresponding chromosomal IS are indicated. Gray areas reflect gaps in sequencing reads related to incomplete sequence coverage. Graphics in **a** and **b** were created using BioRender.com.

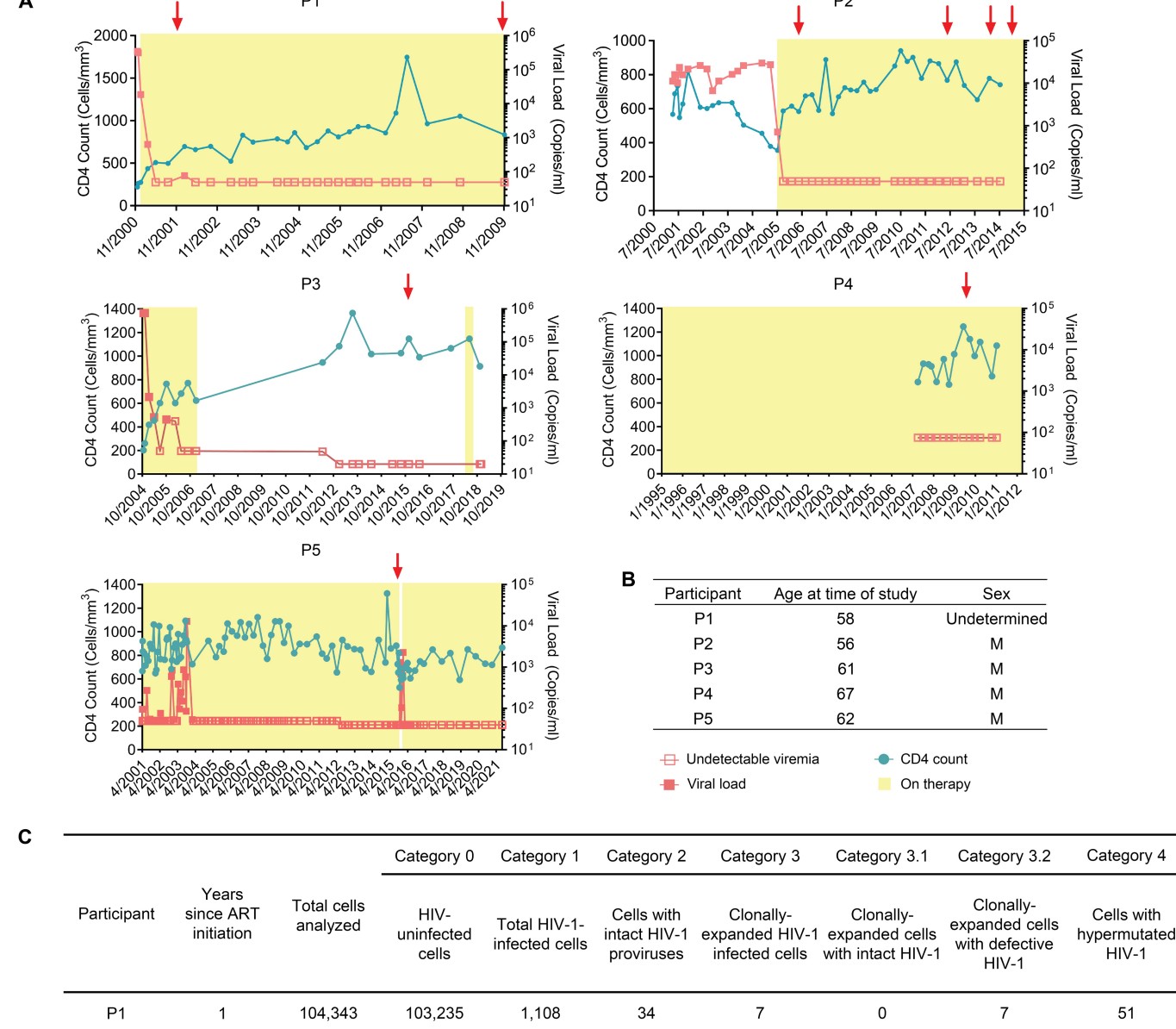

**A**

**B**

| Participant | Age at time of study | Sex |
|---|---|---|
| P1 | 58 | Undetermined |
| P2 | 56 | M |
| P3 | 61 | M |
| P4 | 67 | M |
| P5 | 62 | M |

□ Undetectable viremia  ● CD4 count
■ Viral load  ▨ On therapy

**C**

| Participant | Years since ART initiation | Total cells analyzed | Category 0 HIV-uninfected cells | Category 1 Total HIV-1-infected cells | Category 2 Cells with intact HIV-1 proviruses | Category 3 Clonally-expanded HIV-1 infected cells | Category 3.1 Clonally-expanded cells with intact HIV-1 | Category 3.2 Clonally-expanded cells with defective HIV-1 | Category 4 Cells with hypermutated HIV-1 |
|---|---|---|---|---|---|---|---|---|---|
| P1 | 1 | 104,343 | 103,235 | 1,108 | 34 | 7 | 0 | 7 | 51 |
| P1 | 9 | 169,346 | 169,166 | 180 | 14 | 7 | 3 | 4 | 36 |
| P2 | 1 | 153,231 | 150,183 | 3,048 | 10 | 40 | 0 | 40 | 105 |
| P2 | 8 | 152,892 | 152,439 | 453 | 35 | 18 | 18 | 0 | 15 |
| P3 | 11 | 29,959 | 29,199 | 760 | 34 | 21 | 8 | 13 | 233 |
| P4 | 15 | 102,849 | 102,355 | 494 | 48 | 58 | 14 | 44 | 26 |
| P5 | 14 | 75,097 | 74,125 | 972 | 62 | 21 | 13 | 8 | 88 |
| Total | | 787,717 | 780,702 | 7,015 | 237 | 172 | 56 | 116 | 554 |

**Extended Data Fig. 2 | Clinical and demographical data of study participants donating peripheral blood samples.** (A) summarizes longitudinal CD4 T cell counts and viral loads; red arrows indicate the timepoints of PBMC sample collection. Note that study participant 3 is an elite controller or post-treatment controller. In study participant 4, continuous suppressive antiretroviral therapy since 1995 was documented, but individual viral load test results were not available for review by study team. (B) demonstrates sex and age at the time of study participation. (C): Cell numbers analyzed at each timepoint from each study participant in each cell category.

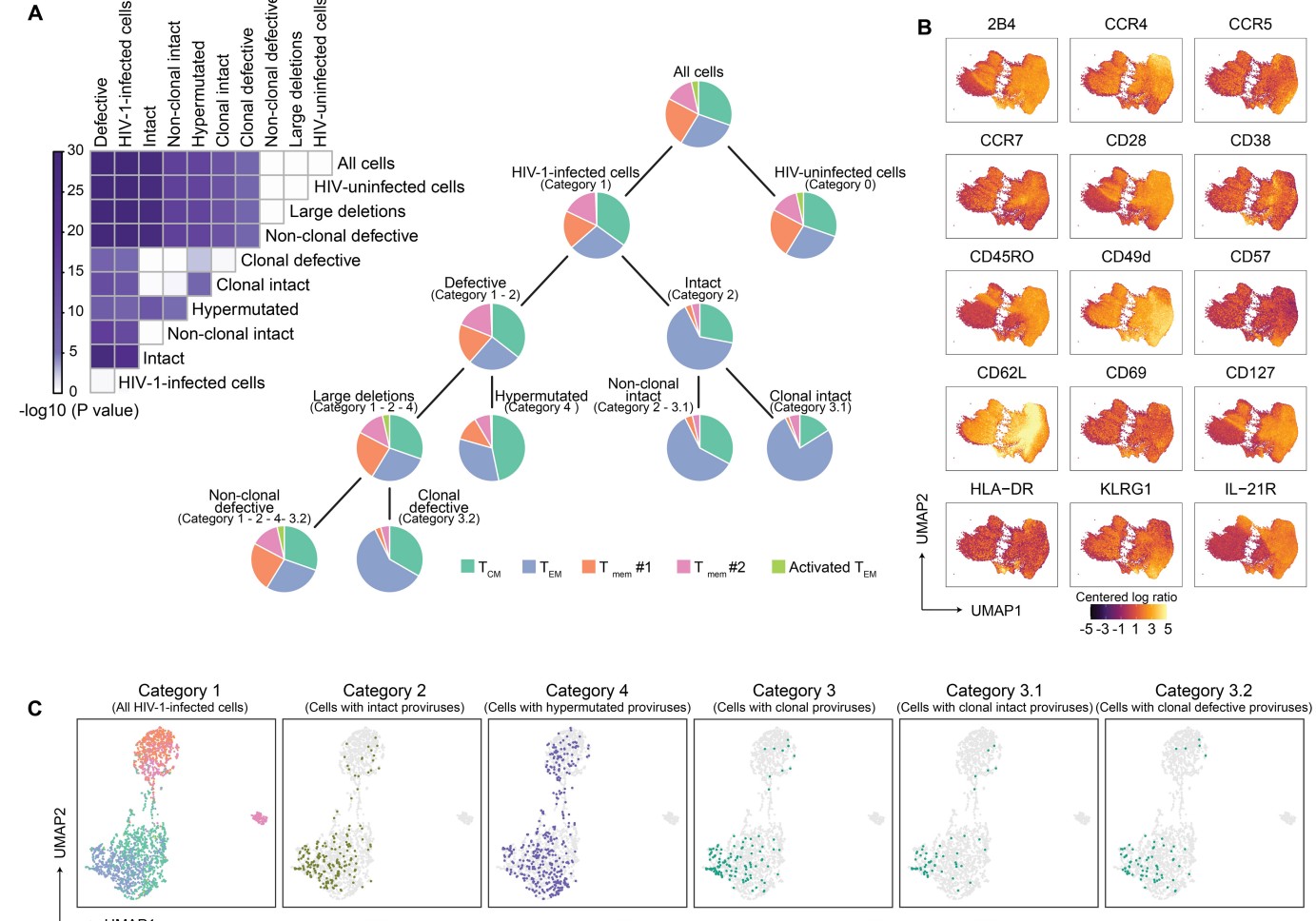

**Extended Data Fig. 3 | Global phenotypic analysis of HIV-1 reservoir cells from peripheral blood.** (A): Pie charts indicating proportional contributions of cells in each of the phenotypic clusters (defined in Fig. 2a) to the total number of cells in indicated cell categories. Heatmap reflects FDR-adjusted p-values from comparisons between individual pie charts, based on two-sided chi-squared tests. (B): UMAP plots displaying the expression of selected surface markers across all five UMAP spherical clusters. (C): UMAP plots generated from all category 1 cells; positioning of category 2-4 cells is indicated individually.

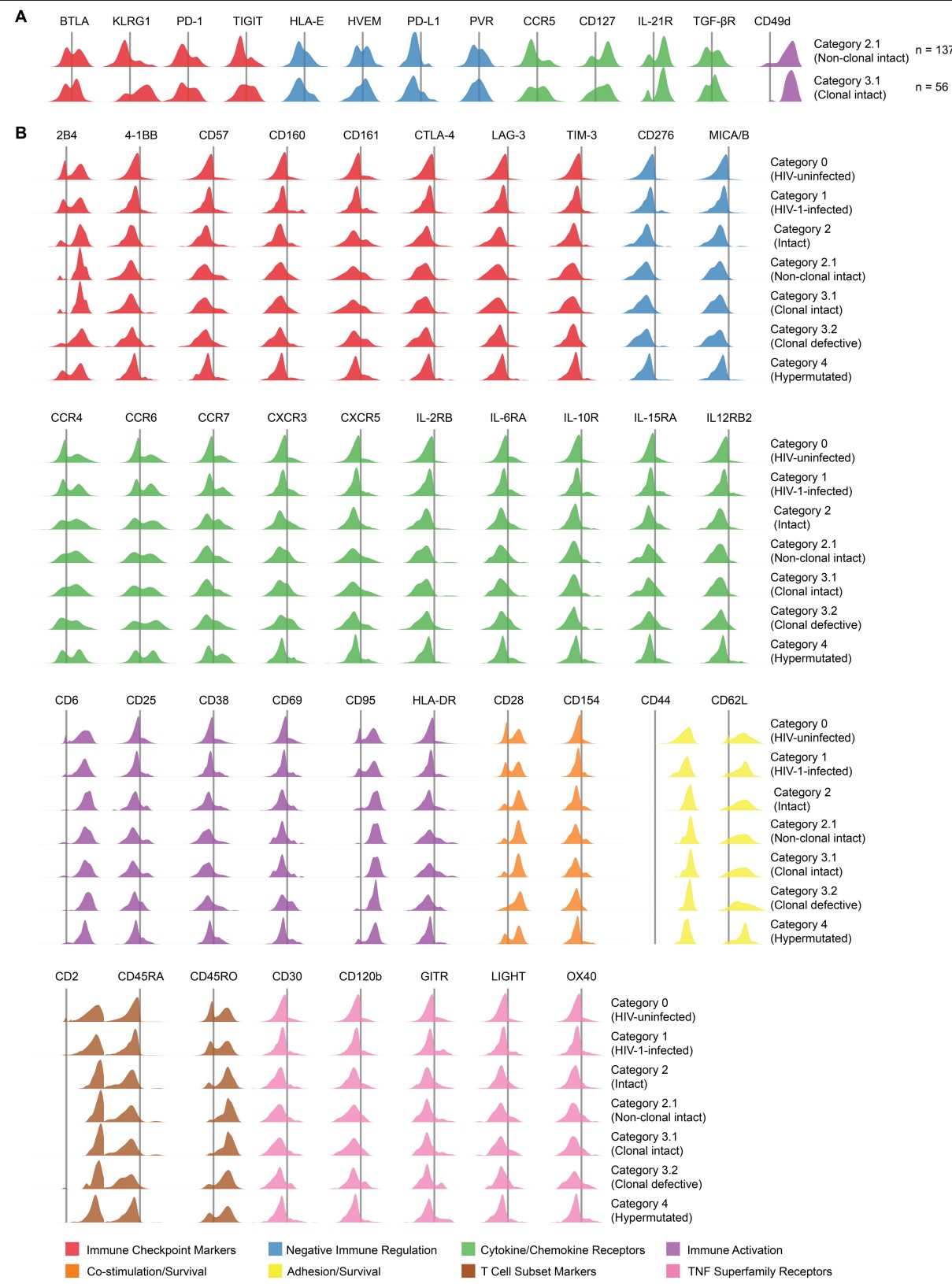

**Extended Data Fig. 4 | Comprehensive phenotypic profile of HIV-1 reservoir cells from peripheral blood.** (A): Density plots reflecting surface expression of selected phenotypic markers in cells harboring intact proviruses detected only once (non-clonal intact, category 2.1) and in cells harboring clonal intact proviruses (clonal intact, category 3.1). (B): Density plots reflecting surface expression of indicated phenotypic markers on HIV-1-infected peripheral blood mCD4⁺ T cells from categories 1-4. Category 0 cells are shown as reference.

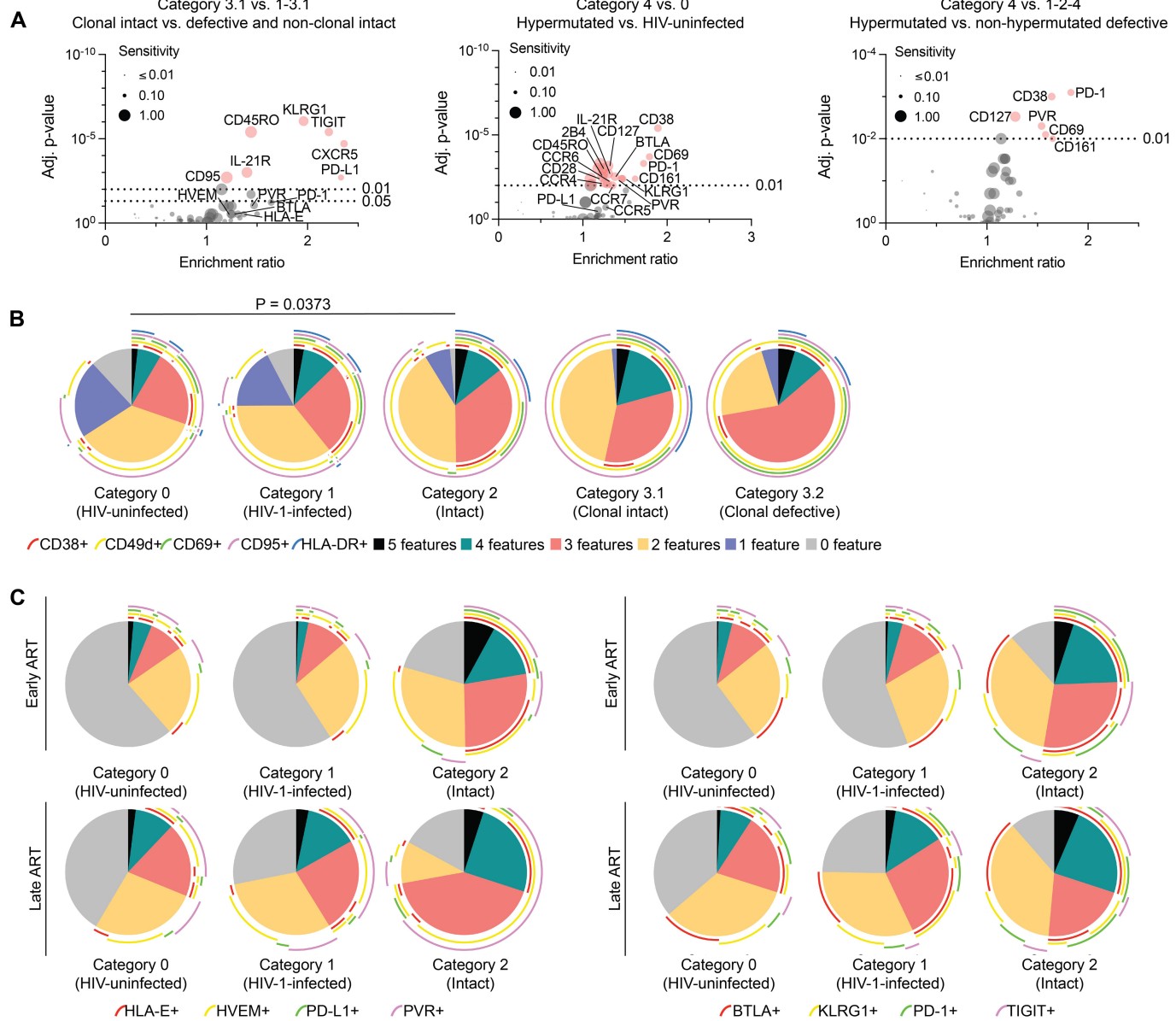

**Extended Data Fig. 5 | Additional phenotypic characteristics of HIV-1 reservoir cells from peripheral blood.** (A): Volcano plots displaying enrichment ratios and corresponding FDR-adjusted p-values for all 53 surface markers included in this study for comparisons between indicated HIV-1 reservoir cell populations. Selected markers are labeled individually. Marker sensitivities are indicated by the sizes of their corresponding dots. Bootstrapped data from comparisons between indicated categories of cells are shown. Significance was tested using a two-sided chi-squared test. (B):

Ensemble expression of immune activation markers on HIV-1 reservoir cells. SPICE diagrams reflect proportions of cells in each category expressing 0, 1, 2, 3, 4 or 5 of the indicated T cell activation markers. Overlaying arches reflect proportions of cells expressing individual markers. (C): Ensemble expression of indicated phenotypic markers on category 0, 1, 2 HIV-1 reservoir cells at early and late timepoints after ART initiation in study participants P1 and P2. (B-C): One diagram is shown for each indicated category of cells. Significance was tested using an FDR-adjusted permutation test.

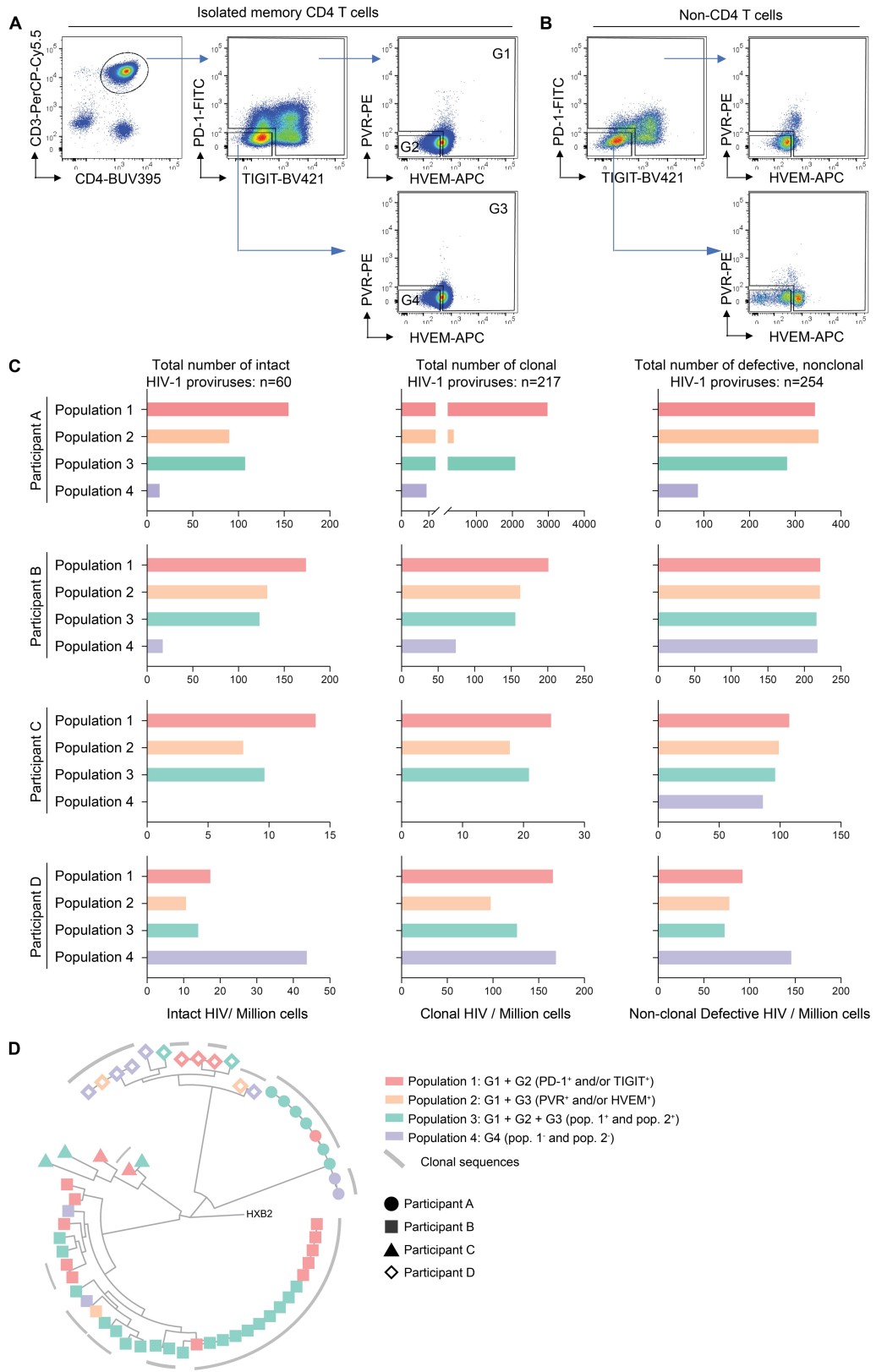

**Extended Data Fig. 6 | Orthogonal validation of phenotypic signatures of HIV-1 reservoir cells using fluorescence-activated cell sorting of patient-derived memory CD4+ T lymphocytes.** (A): Representative flow cytometry dot plots demonstrating the gating strategy for mCD4+ T cell populations isolated from peripheral blood; gates used for sorting (G1-G4) are indicated. (B): Flow cytometry plots for displaying expression of indicated markers on non-CD4+ T cells are shown for comparative purposes. (C): Frequencies of intact proviruses, sequence-identical clonal proviruses (including both intact and defective proviruses) and non-clonal defective proviruses in indicated, sorted mCD4+ T cell subpopulations, determined by near full-length proviral sequencing (FLIP-seq). (D): Phylogenetic tree of all intact proviruses retrieved from sorted mCD4+ T cell subpopulations. Note that in study participant D, the majority of intact proviruses were detected within mCD4+ T cells that were negative for PD-1, TIGIT, HVEM and PVR but positive for BTLA/KLRG-1 and/or HLA-E/PD-L1.

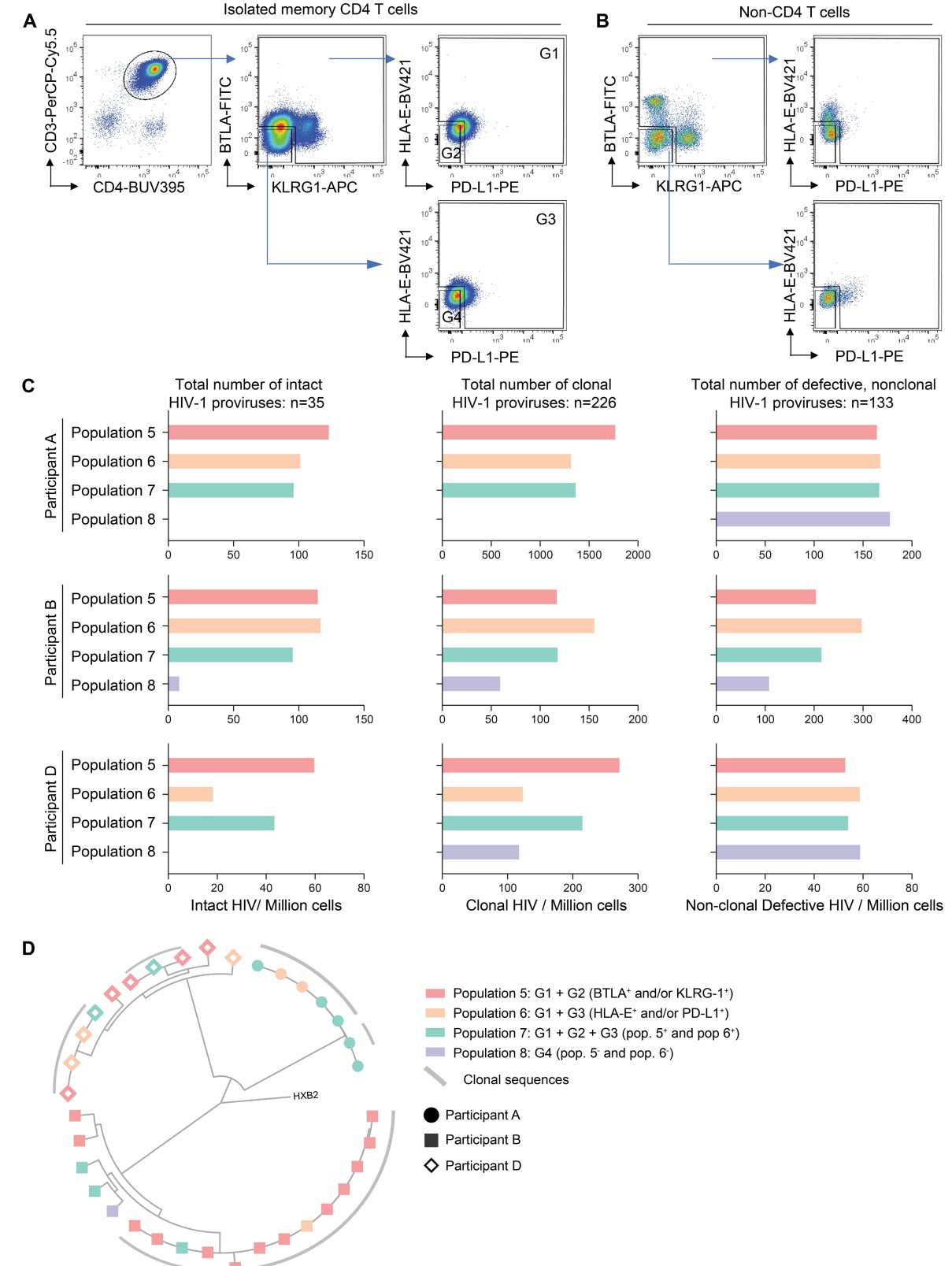

**Extended Data Fig. 7 | Orthogonal validation of phenotypic signatures of HIV-1 reservoir cells using fluorescence-activated cell sorting of patient-derived memory CD4⁺ T lymphocytes.** (A): Representative flow cytometry dot plots demonstrating the gating strategy for memory CD4⁺ T cell populations isolated from peripheral blood; gates used for sorting (G1-G4) are indicated. (B): Flow cytometry plots for displaying expression of indicated markers on non-CD4⁺ T cells are shown for comparative purposes. (C): Frequencies of intact proviruses, sequence-identical clonal proviruses (including both intact and defective proviruses) and non-clonal defective proviruses in indicated, sorted memory CD4⁺ T cell subpopulations, determined by near full-length proviral sequencing (FLIP-seq). (D): Phylogenetic tree of all intact proviruses retrieved from sorted mCD4⁺ T cell subpopulations. Note that in study participant D, the majority of intact proviruses were detected within memory CD4⁺ T cells that were negative for PD-1, TIGIT, HVEM and PVR but positive for BTLA/KLRG-1 and/or HLA-E/PD-L1.

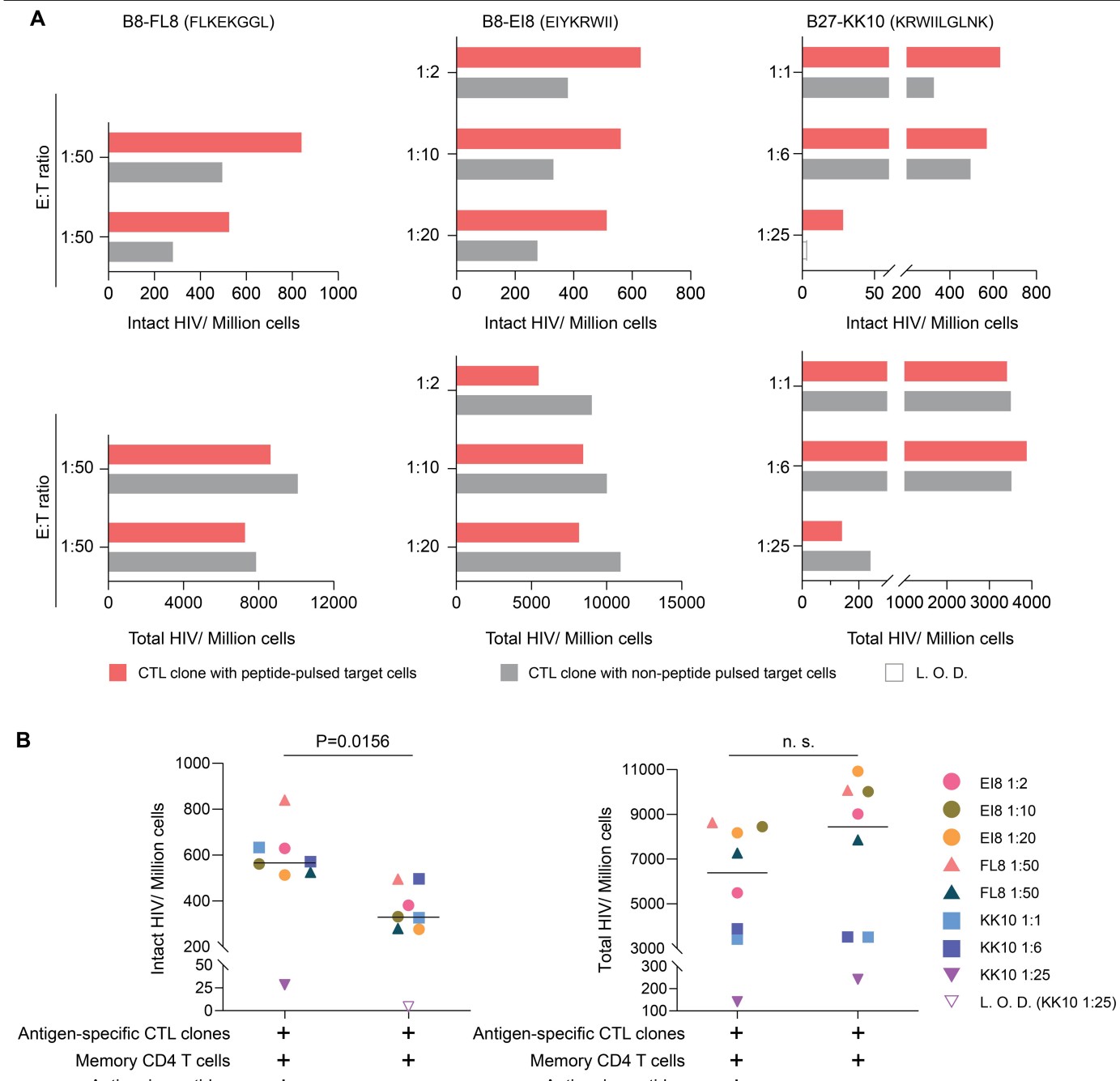

**Extended Data Fig. 8 | Evaluation of the susceptibility of HIV-1 reservoir cells to cytotoxic effects of HIV-1-specific CD8[+] T cell clones.** (A): Patient-derived memory CD4[+] T cells were pulsed with indicated epitopic peptides, followed by exposure to HIV-1-specific CD8[+] T cell clones matched to the indicated autologous HLA class I alleles and recognizing the epitopic peptides used for target cell pulsing. Following incubation for 16 h, viable CD3[+]CD4[+] T cells were sorted and subjected to assessments of intact proviruses using the IPDA. Data indicate frequencies of intact proviruses in patient-derived cells, relative to autologous control cells incubated with the same antigen-specific

CD8[+] T cell clones in the absence of prior peptide pulsing. Frequencies of total HIV-1 DNA in sorted cells, evaluated by the IPDA, are shown for comparison. Experiments were conducted at indicated effector-to-target cell ratios, taking into consideration the CTL clone and target cell availability and the killing potency of the CTL clones. L. O. D. Limit of detection, reflecting the number of cells analyzed without target identification. (B): Summary of data shown in A. Geometric symbols reflect individual CTL and target cell pairs. p-value adjusted for multiple comparison was determined by two-sided Wilcoxon matched-pairs signed rank test.

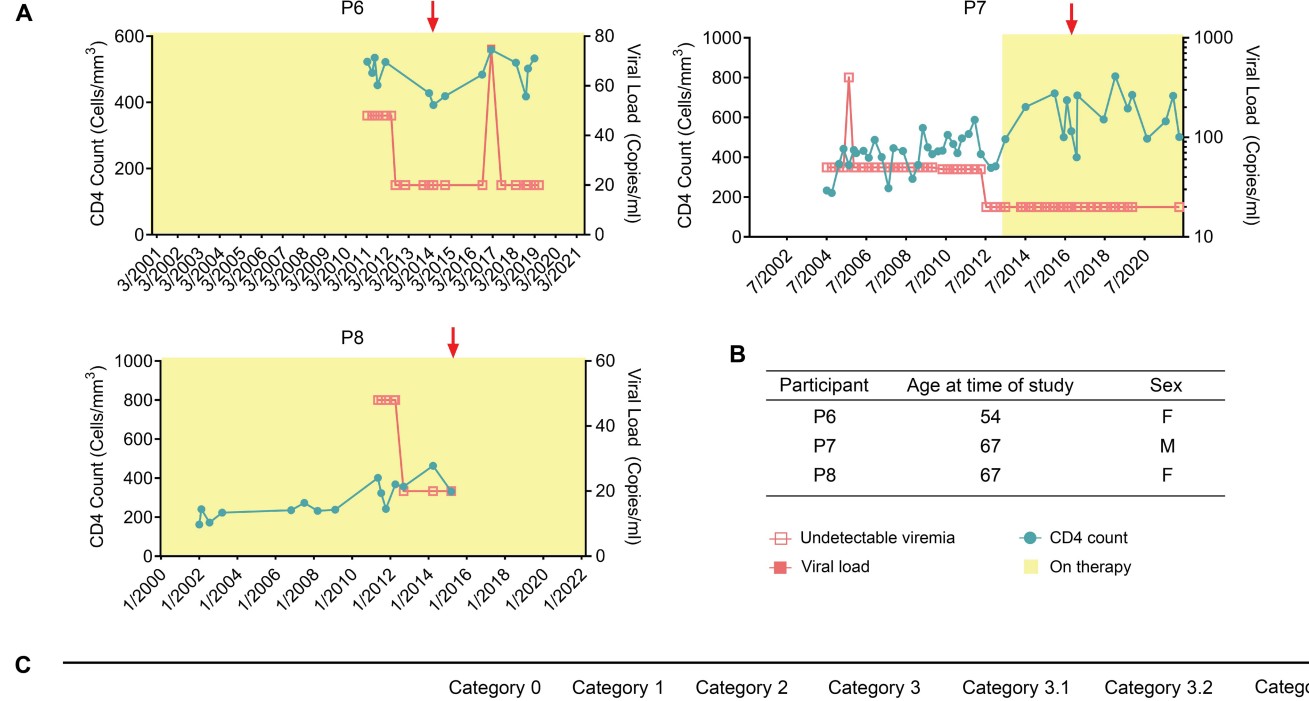

**A**

**B**

| Participant | Age at time of study | Sex |
|---|---|---|
| P6 | 54 | F |
| P7 | 67 | M |
| P8 | 67 | F |

□ Undetectable viremia ● CD4 count
■ Viral load ▨ On therapy

**C**

| Participant | Years since ART initiation | Total cells analyzed | Category 0 HIV-uninfected cells | Category 1 Total HIV-1-infected cells | Category 2 Cells with intact HIV-1 proviruses | Category 3 Clonally-expanded HIV-1 infected cells | Category 3.1 Clonally-expanded cells with intact HIV-1 | Category 3.2 Clonally-expanded cells with defective HIV-1 | Category 4 Cells with hypermutated HIV-1 |
|---|---|---|---|---|---|---|---|---|---|
| P6 | 13 | 96,855 | 93,716 | 3,139 | 94 | 2 | 0 | 2 | 13 |
| P7 | 12 | 153,766 | 153,682 | 84 | 10 | 7 | 0 | 7 | 25 |
| P8 | 15 | 146,007 | 145,342 | 665 | 7 | 0 | 0 | 0 | 1 |
| Total | | 396,628 | 392,740 | 3,888 | 111 | 9 | 0 | 9 | 39 |

**Extended Data Fig. 9 | Clinical and demographical data of study participants donating lymph node samples.** (A) summarizes longitudinal CD4⁺ T cell counts and viral loads; red arrows indicate the timepoints of LN sample collection. In study participants P6 and P8, continuous suppressive antiretroviral therapy since 2001/2002 was documented, but individual viral load test results were not available for review by study team. In study participant P7, undetectable plasma viremia was documented since 2004, but clinical records summarizing antiretroviral treatment history prior to 2012 were not available to study team. (B) demonstrates sex and age at the time of study participation. (C): Cell numbers analyzed from each study participant in each cell category.

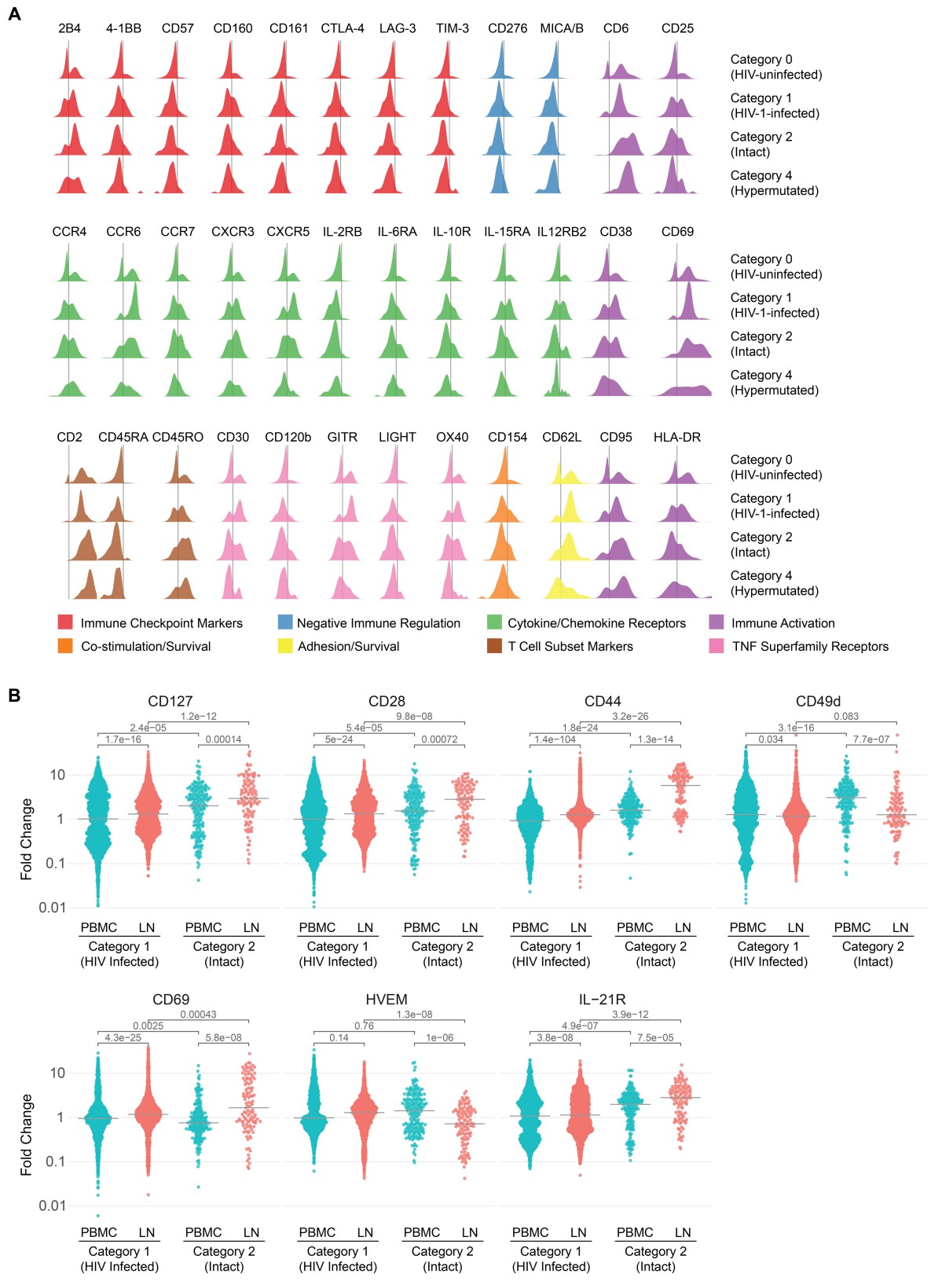

**Extended Data Fig. 10 | Additional phenotypic properties of HIV-1 reservoir cells from lymph nodes.** (A): Density plots reflecting the surface expression of indicated phenotypic markers on HIV-1-infected mCD4⁺ T cells in lymph nodes from categories 1, 2 and 4. Category 0 cells are shown for comparison. (B): Exploratory comparison of selected phenotypic markers between reservoir cells from peripheral blood and lymph nodes. Data reflect fold-changes in expression intensities of indicated surface markers in category 1 cells (all HIV-1-infected cells) and category 2 cells (enriched for cells harboring intact HIV-1 proviruses), normalized to category 0 cells (uninfected cells), from peripheral blood and lymph nodes. Significance was tested using two-sided Mann Whitney U tests, FDR-adjusted p-values are shown.

# Reporting Summary

## Statistics

For all statistical analyses, confirm that the following items are present in the figure legend, table legend, main text, or Methods section.

| n/a | Confirmed | |
|---|---|---|
| ☐ | ☒ | The exact sample size (*n*) for each experimental group/condition, given as a discrete number and unit of measurement |
| ☐ | ☒ | A statement on whether measurements were taken from distinct samples or whether the same sample was measured repeatedly |
| ☐ | ☒ | The statistical test(s) used AND whether they are one- or two-sided *Only common tests should be described solely by name; describe more complex techniques in the Methods section.* |
| ☒ | ☐ | A description of all covariates tested |
| ☐ | ☒ | A description of any assumptions or corrections, such as tests of normality and adjustment for multiple comparisons |
| ☐ | ☒ | A full description of the statistical parameters including central tendency (e.g. means) or other basic estimates (e.g. regression coefficient) AND variation (e.g. standard deviation) or associated estimates of uncertainty (e.g. confidence intervals) |
| ☐ | ☒ | For null hypothesis testing, the test statistic (e.g. $F$, $t$, $r$) with confidence intervals, effect sizes, degrees of freedom and $P$ value noted *Give P values as exact values whenever suitable.* |
| ☒ | ☐ | For Bayesian analysis, information on the choice of priors and Markov chain Monte Carlo settings |
| ☒ | ☐ | For hierarchical and complex designs, identification of the appropriate level for tests and full reporting of outcomes |
| ☒ | ☐ | Estimates of effect sizes (e.g. Cohen's *d*, Pearson's *r*), indicating how they were calculated |

*Our web collection on statistics for biologists contains articles on many of the points above.*

## Software and code

Policy information about availability of computer code

| | |
|---|---|
| Data collection | Tapestri (MissionBio, v2.0.1) |
| Data analysis | Bioconductor (https://bioconductor.org/, version 3.14), Biorender (https://biorender.com), Cutadapt (https://cutadapt.readthedocs.io/en/stable/, version 2.5), Graphpad prism (https://www.graphpad.com/scientific-software/prism/, version 8.2.1), Highlighter (https://www.hiv.lanl.gov/content/sequence/HIGHLIGHT/highlighter_top.html), MAFFT (https://mafft.cbrc.jp/alignment/software, version 7), MEGA (https://www.megasoftware.net, version 7.0.26), MUSCLE (http://www.drive5.com/muscle, version 3.8.1551), Python (https://www.python.org, version 3.6), R (https://www.r-project.org, version 4.1), SPICE (https://niaid.github.io/spice/, version 6.0), bwa (https://github.com/lh3/bwa, version 0.7.12), monocle3 (https://cole-trapnell-lab.github.io/monocle3/, version 1.0.0), samtools and bcftools (http://www.htslib.org/, version 1.9), Adobe Illustrator (https://www.adobe.com/products/illustrator.html, version 2022 (26.1)), Flow Jo (V10.7), DiVa (V08.01), Leiden Community detection algorithm from Monocle v3 (https://cole-trapnell-lab.github.io/monocle3/, version 1.0.0) |

For manuscripts utilizing custom algorithms or software that are central to the research but not yet described in published literature, software must be made available to editors and reviewers. We strongly encourage code deposition in a community repository (e.g. GitHub). See the Nature Portfolio guidelines for submitting code & software for further information.

## Data

Policy information about availability of data

All manuscripts must include a data availability statement. This statement should provide the following information, where applicable:

- Accession codes, unique identifiers, or web links for publicly available datasets
- A description of any restrictions on data availability
- For clinical datasets or third party data, please ensure that the statement adheres to our policy

Owing to study participant confidentiality concerns, viral sequencing data cannot be publicly released, but will be made available to investigators upon reasonable request and after signing a data sharing agreement. Correspondence and requests for materials should be addressed to Dr. Mathias Lichterfeld (mlichterfeld@partners.org).

# Field-specific reporting

Please select the one below that is the best fit for your research. If you are not sure, read the appropriate sections before making your selection.

☒ Life sciences          ☐ Behavioural & social sciences          ☐ Ecological, evolutionary & environmental sciences

For a reference copy of the document with all sections, see nature.com/documents/nr-reporting-summary-flat.pdf

# Life sciences study design

All studies must disclose on these points even when the disclosure is negative.

| | |
|---|---|
| Sample size | A total of n=4 ART-treated participants, and n=1 EC were analyzed in data described in Figure 1-4 (peripheral blood viral reservoir cells). Data from n=3 study participants are described in Figure 5 (lymph node reservoir cells). No computational approach was used to determine these sample sizes, testing was based on availability of more than 1 million memory CD4 T cells per study participant. |
| Data exclusions | No data from the described individuals were excluded. |
| Replication | Multiple microfluidic cartridges were run for each patient sample. In selected cases (n=14), the proviral library from a given cartridge was sequenced twice, with similar results. |
| Randomization | No randomization was performed, because we performed an analysis of study participants enrolled in an observational study. |
| Blinding | Coded samples from study participants were used throughout the study; laboratory personnel was not blinded with regard to the respective study subjects, since this was a non-interventional, observational study. |

# Reporting for specific materials, systems and methods

We require information from authors about some types of materials, experimental systems and methods used in many studies. Here, indicate whether each material, system or method listed is relevant to your study. If you are not sure if a list item applies to your research, read the appropriate section before selecting a response.

## Materials & experimental systems

| n/a | Involved in the study |
|---|---|
| ☐ | ☒ Antibodies |
| ☒ | ☐ Eukaryotic cell lines |
| ☒ | ☐ Palaeontology and archaeology |
| ☒ | ☐ Animals and other organisms |
| ☐ | ☒ Human research participants |
| ☒ | ☐ Clinical data |
| ☒ | ☐ Dual use research of concern |

## Methods

| n/a | Involved in the study |
|---|---|
| ☒ | ☐ ChIP-seq |
| ☐ | ☒ Flow cytometry |
| ☒ | ☐ MRI-based neuroimaging |

## Antibodies

| | |
|---|---|
| Antibodies used | All antibodies were purchased from Biolegend, BD and eBioscience. All antibodies and clone names are provided in the methods section of the manuscript. A cocktail of lyophilized antibodies from Biolegend was used for proteogenomic profiling experiments which contained the following antibodies: PD-L1 (clone 29E.2A3), CD276 (clone DCN.70), HVEM (clone122), CD155 (clone SKII.4), CD154/CD40L (clone 24-31), CCR4 (clone L291H4), PD-1 (A17188B), TIGIT (clone A15153G), CD44 (clone BJ18), CXCR3 (clone G025H7), CCR5 (clone HEK/1/85a), CCR6 (clone G034E3), CXCR5 (clone J252D4), CCR7 (clone G043H7), KLRB1/CD161 (clone HP-3G10), CTLA-4 (clone BNI3), LAG-3 (clone 11C3C65), KLRG1 (clone 14C2A07), CD95 (clone, DX2), OX40/CD134 (clone Ber-ACT35), CD57 (clone HNK-1), TIM-3 (clone F38-2E2), BTLA/CD272 (clone MIH26), CD244/2B4 (clone 2-69), IL-2R (clone TU27), CD137/4-1BB |

(clone 4B4-1), GITR/CD357 (clone 108-17), CD28 (clone CD28.2), CD127 (clone A019D5), IL-6R (clone UV4), HLA-E (clone 3D12), MICA/B (clone 6D4), IL-15R/CD215 (clone JM7A4), IL-21R (clone 2G1-K12), TNFR2 (clone 3G7A02), CD160 (clone BY55), LIGHT/CD258 (clone T5-39), IL-10R/CD210 (clone 3F9), TGFB-R (clone W17055E), IL-12R (clone S16020B), CD6 (clone BL-CD6), CD49d (clone 9F10), CD25 (clone BC96), CD30 (clone BY88), CD69 (clone FN50), CD45RA (clone HI100), CD38 (clone HIT2), HLA-DR (clone L243), CD4 (clone RPA-T4), CD2 (clone TS1/8), CD3 (clone UCHT1), CD62L (clone DREG-56), CD45RO (clone UCHL1). IgG2a (Biolegend, clone MOPC-173, catalog 400299), IgG2b (Biolegend, clone MPC-11, catalog 400383).

The following antibodies were used for cell sorting: CD3-PerCP-Cy5.5 (BioLegend, clone UCHT1, catalog 300430), CD3-FITC (BioLegend, clone UCHT1, catalog 300406), CD4-BUV395 (BD, clone RPA-T4, catalog 564724), CD4-BV711 (BioLegend, clone RPA-T4, catalog 300558), PD1-FITC (BioLegend, clone A17188B, catalog 621612), TIGIT-BV421 (BioLegend, clone A15153G, catalog 372710), PVR-PE (eBioscience, clone 2H7CD155, catalog 12-1550-41), HVEM-APC (Biolegend, clone 122, catalog 318808), BTLA-FITC (Biolegend, clone MIH26, catalog 344524), KLRG1-APC (Biolegend, clone 14C2A07, catalog 368606), HLAE-BV421 (Biolegend, clone 3D12, catalog 342612), PDL1-PE (Biolegend, clone MIH2, catalog 393608)

| Validation | All antibodies were validated by manufacturers. Validation information is available for each antibody at www.biolegend.com, https://www.bd.com/en-us, and https://www.thermofisher.com/us/en/home/life-science/antibodies/ebioscience.html. |
|---|---|

The lyophilized antibody cocktail from Biolegend was validated by Biolegend.

CD3-PerCP-Cy5.5 (BioLegend, clone UCHT1, catalog 300430)
Reactivity: Human
Host Species: Mouse
Application: FC - Quality tested
Application Notes: Additional reported applications (for the relevant formats) include: immunohistochemical staining of acetone-fixed frozen sections4,6,7 and formalin-fixed paraffin-embedded sections11, immunoprecipitation1, activation of T cells2,3,5, Western blotting9, and spatial biology (IBEX)16,17. The LEAF™ purified antibody (Endotoxin < 0.1 EU/μg, Azide-Free, 0.2 μm filtered) is recommended for functional assays (Cat. No. 300413, 300414, and 300432). For highly sensitive assays, we recommend Ultra-LEAF™ purified antibody (Cat. No. 300437, 300438, 300465, 300466, 300473, 300474) with a lower endotoxin limit than standard LEAF™ purified antibodies (Endotoxin < 0.01 EU/μg).
Application References: Salmeron A, et al. 1991. J. Immunol. 147:3047. (IP)
Graves J, et al. 1991. J. Immunol. 146:2102. (Activ)
Lafont V, et al. 2000. J. Biol. Chem. 275:19282. (Activ)
Ryschich E, et al. 2003. Tissue Antigens 62:48. (IHC)
Thompson AG, et al. 2004. J. Immunol. 173:1671. (Activ)
Sakkas LI, et al. 1998. Clin. Diagn. Lab. Immun. 5:430. (IHC)
Mack CL, et al. 2004. Pediatr. Res. 56:79. (IHC)
Thakral D, et al. 2008. J. Immunol. 180:7431. (FC) PubMed
Van Dongen JJM, et al. 1988. Blood 71:603. (WB)
Yoshino N, et al. 2000. Exp. Anim. (Tokyo) 49:97. (FC)
Pollard, K. et al. 1987. J. Histochem. Cytochem. 35:1329. (IHC)
Luckashenak N, et al. 2013. J. Immunol. 190:27. PubMed

CD3-FITC (BioLegend, clone UCHT1, catalog 300406)
Reactivity: Human
Host Species: Mouse
Application: FC - Quality tested
Application Notes: Additional reported applications (for the relevant formats) include: immunohistochemical staining of acetone-fixed frozen sections4,6,7 and formalin-fixed paraffin-embedded sections11, immunoprecipitation1, activation of T cells2,3,5, Western blotting9, and spatial biology (IBEX)16,17. The LEAF™ purified antibody (Endotoxin < 0.1 EU/μg, Azide-Free, 0.2 μm filtered) is recommended for functional assays (Cat. No. 300413, 300414, and 300432). For highly sensitive assays, we recommend Ultra-LEAF™ purified antibody (Cat. No. 300437, 300438, 300465, 300466, 300473, 300474) with a lower endotoxin limit than standard LEAF™ purified antibodies (Endotoxin < 0.01 EU/μg).
Application References: Salmeron A, et al. 1991. J. Immunol. 147:3047. (IP)
Graves J, et al. 1991. J. Immunol. 146:2102. (Activ)
Lafont V, et al. 2000. J. Biol. Chem. 275:19282. (Activ)
Ryschich E, et al. 2003. Tissue Antigens 62:48. (IHC)
Thompson AG, et al. 2004. J. Immunol. 173:1671. (Activ)
Sakkas LI, et al. 1998. Clin. Diagn. Lab. Immun. 5:430. (IHC)
Mack CL, et al. 2004. Pediatr. Res. 56:79. (IHC)
Thakral D, et al. 2008. J. Immunol. 180:7431. (FC) PubMed
Van Dongen JJM, et al. 1988. Blood 71:603. (WB)
Yoshino N, et al. 2000. Exp. Anim. (Tokyo) 49:97. (FC)
Pollard, K. et al. 1987. J. Histochem. Cytochem. 35:1329. (IHC)
Luckashenak N, et al. 2013. J. Immunol. 190:27. PubMed

CD4-BUV395 (BD, clone RPA-T4, catalog 564724)
Reactivity: Human (QC Testing)
Host Species: Mouse
Application: Flow cytometry (Routinely Tested)

CD4-BV711 (BioLegend, clone RPA-T4, catalog 300558)
Reactivity: Human
Host Species: Mouse
Application: FC - Quality tested
Application Notes: The RPA-T4 antibody binds to the D1 domain of CD4 (CDR1 and CDR3 epitopes) and can block HIV gp120 binding and inhibit syncytia formation. Additional reported applications (for the relevant formats) include: immunohistochemistry of

acetone-fixed frozen sections3,4,5, blocking of T cell activation1,2, and spatial biology (IBEX)10,11. This clone was tested in-house and does not work on formalin fixed paraffin-embedded (FFPE) tissue. The Ultra-LEAF™ purified antibody (Endotoxin < 0.01 EU/μg, Azide-Free, 0.2 μm filtered) is recommended for functional assays (Cat. No. 300569 - 300574).

Application References: Knapp W, et al. 1989. Leucocyte Typing IV. Oxford University Press. New York.
Moir S, et al. 1999. J. Virol. 73:7972. (Activ)
Deng MC, et al. 1995. Circulation 91:1647. (IHC)
Friedman T, et al. 1999. J. Immunol. 162:5256. (IHC)
Mack CL, et al. 2004. Pediatr. Res. 56:79. (IHC)
Lan RY, et al. 2006. Hepatology 43:729.
Zenaro E, et al. 2009. J. Leukoc. Biol. 86:1393. (FC) PubMed
Yoshino N, et al. 2000. Exp. Anim. (Tokyo) 49:97. (FC)
Stoeckius M, et al. 2017. Nat. Methods. 14:865. (PG)
Radtke AJ, et al. 2020. Proc Natl Acad Sci USA. 117:33455-33465. (SB) PubMed
Radtke AJ, et al. 2022. Nat Protoc. 17:378-401. (SB) PubMed

PD1-FITC (BioLegend, clone A17188B, catalog 621612)
Reactivity: Human
Host Species: Mouse
Application: FC - Quality tested
Application Notes: A17188B antibody can block the binding of NAT105 and EH12.2H7 antibodies to the target.

TIGIT-BV421 (BioLegend, clone A15153G, catalog 372710)
Reactivity: Human
Host Species: Mouse
Application: FC - Quality tested
Application Notes: This clone can suppress anti-CD3 induced T cell proliferation in vitro based on in-house testing. This clone has been tested in-house and determined to not be suitable for applications in immunohistochemistry of paraffin-embedded tissue sections (IHC-P).
Additional reported applications (for the relevant formats) include: Blocking
Application References: Stamm H, et al. 2018. Oncogene. Pubmed

PVR-PE (eBioscience, clone 2H7CD155, catalog 12-1550-41)
Reactivity: Human
Host Species: Mouse
Application: Flow Cytometry (Flow)

HVEM-APC (Biolegend, clone 122, catalog 318808)
Reactivity: Human
Host Species: Mouse
Application: FC - Quality tested
Application Notes: The 122 antibody has been shown to be useful for flow cytometry, Western blot, and ELISA.
Application References: Cheung TC, et al. 2010. J. Immunol. 185:1949. PubMed
Hobo W, et al. 2012. J Immunol. 189:39. PubMed.

BTLA-FITC (Biolegend, clone MIH26, catalog 344524)
Reactivity: Human
Host Species: Mouse
Application: FC - Quality tested
Application Notes: Additional reported applications (for the relevant formats) include: inhibition of T cell proliferation and cytokine production1. Clone MIH26 has agonistic activity on BTLA, resulting in the inhibition of activation.
Application References: Otsuki N, et al. 2006. Biochem. Bioph. Res. Co. 344:1121.
Okano M, et al. 2008. Clin. Exp. Allergy 38:1891.

KLRG1-APC (Biolegend, clone 14C2A07, catalog 368606)
Reactivity: Human
Host Species: Mouse
Application: FC - Quality tested

HLAE-BV421 (Biolegend, clone 3D12, catalog 342612)
Reactivity: Human
Host Species: Mouse
Application: FC - Quality tested
Application References: Lee N, et al. 1998. Proc. Natl. Acad. Sci. USA. 95:5199.
Wooden SL, et al. 2005. J. Immunol. 175:1383.
Monaco EL, et al. 2008. J. Immunol. 181:5442.
Corrah TW, et al. 2011. J. Virol. 85:3367.

PDL1-PE (Biolegend, clone MIH2, catalog 393608)
Reactivity: Human
Host Species: Mouse
Application: FC - Quality tested

# Human research participants

Policy information about studies involving human research participants

| | |
|---|---|
| Population characteristics | Clinical characteristics of the study participants are shown in the Extended Data Figure 1A-B and the Extended Data Figure 9A-B |
| Recruitment | All study persons were recruited based on referral by HIV clinicians and infectious disease physicians. The enrollment protocols allowed recruited of men and women >18 years old, of any race or ethnicity. Patients were included in our prior studies and selected for this project according to the following criteria: availability of sufficient cells for experiments, availability of full-genome sequencing data and proviral integration site data, and relatively high frequency of genome-intact HIV-1 proviruses in CD4 T cells. |
| Ethics oversight | The MassGeneralBrigham Human Research Committee approved all sample collection at MGH and BWH; the IRB of the NIH supervised sample collection at the NIH Clinical Center. |

Note that full information on the approval of the study protocol must also be provided in the manuscript.

# Flow Cytometry

## Plots

Confirm that:

☒ The axis labels state the marker and fluorochrome used (e.g. CD4-FITC).

☒ The axis scales are clearly visible. Include numbers along axes only for bottom left plot of group (a 'group' is an analysis of identical markers).

☒ All plots are contour plots with outliers or pseudocolor plots.

☒ A numerical value for number of cells or percentage (with statistics) is provided.

## Methodology

| | |
|---|---|
| Sample preparation | As described in the methods section of the manuscript. |
| Instrument | FACS Aria cell sorting device |
| Software | DiVa version V08.0l<br>FlowJo Version Vl0.7 |
| Cell population abundance | Purity of sorted cell populations was >90%. |
| Gating strategy | The lymphocyte population was identified based on FSC/SSC characteristics, followed by identification of singlets on an FSC-Area vs FSC-Height plot. Viable cells were identified using live/dead viabiliy dye. The remaining gating strategy is shown in Extended Data Figure 6 and 7. |

☒ Tick this box to confirm that a figure exemplifying the gating strategy is provided in the Supplementary Information.

