## [Peer Review File · Nature]

Manuscript Title: Phenotypic signs of immune selection in HIV-1 reservoir cells

Reviewer Comments & Author Rebuttals

Reviewer Reports on the Initial Version:

Referees' comments:

Referee #1 (Remarks to the Author):

The authors use novel single-cell, next-generation sequencing technology, which they combine with ex vivo phenotypic profiling of individual HIV-1-infected memory CD4 T cells to study the phenotype of latent HIV-infected cells.

This is one of the key questions in HIV biology and therefore of utmost importance and significance.

They find that cells harboring defective and therefore non-productive HIV-1 proviruses do not exhibit phenotypic differences compared to autologous uninfected cells. They found that phenotypic differences were more apparent for cells infected with HIV-1 genome-intact proviruses. Their overall conclusion seems to be that the changes in expression of cellular genes they observe are more likely to represent a consequence of immune selection i.e. these changes promote the preferential persistence of HIV-1 virally-infected cells with a phenotypic profile optimized towards minimizing exposure to and killing by host immune effector cells.

On the positive side their combination of single-cell, next-generation sequencing combined with ex vivo phenotyping of individual HIV-1-infected memory CD4 T cells describes the application of a nice technology for investigating the phenotype of latently- infected cells. Furthermore, they show differences between the cells harboring intact and incomplete HIV proviruses, and suggest some surface markers (Fig 3) and combinations of surface markers (Fig 4) which are differentially expressed. However, the differences seem to be very small, and (as they correctly state), the populations are obviously heterogeneous.

However, the overall interpretation is that the work is a "negative result" i.e. there is no really distinctive phenotype of latently infected cells with (or at least likely to have) intact HIV proviruses.

The question is therefore whether it's a sufficiently important negative result to justify publication in Nature - or whether, for instance, it's what everybody presumed to be the case anyway – which is perhaps more likely? Their results are obviously constrained by their choice of ~50 antibodies for cell phenotyping i.e. they could have missed interesting markers or phenotypes by restricting themselves to this panel (compared with the 500 - 1000, or more, cell surface proteins present on the cells).

Points:

The manuscript might be made more accessible to the reader if the authors helped their audience:

e.g. "...using a combination of X surface markers, we were able to identify Y% of the latently infected cells. Enrichment based on these markers suggests that 1/Z00 of these cells harbored intact proviruses..." (where 1/Z00 could be 1/100, 1/1,000 etc)...this would help the reader, and importantly they would then be able to validate their findings in independent experiments on different donors.

Maybe the data has not been reported in this way, or validation experiment attempted, because it's just not possible i.e. no really "useful" surface markers were identified... (I note that the phenotypic signatures discussed are extremely general).

They don't provide any orthogonal validation for their data...

Overall, it could (and as presented should) be interpreted as a "negative result" i.e. no really distinctive phenotype of latently infected cells with (or rather, likely to have) intact HIV proviruses.

Other points

They only analyse memory T cells (after magnetic enrichment). Some reports suggest that HIV proviruses may also be found in naive T cells (potentially accounting for a substantial minority of the total reservoir). It would therefore be interesting for them to discuss the pros and cons of this initial magnetic enrichment;

-they seem to find intact HIV proviruses in cells with a mainly effector memory phenotype e.g. Fig 2a and text. Were they surprised by this as central memory T cells are implicated in the HIV reservoir - which makes sense, because they're the longest-lived T cells – this isn't discussed?

-it would be useful to comment on the fraction of the HIV reservoir predicted to be found in peripheral blood cells, rather than secondary lymphoid tissue..... what implications does that have for their findings, which are all based on peripheral blood?

Is it correct that the volcano plots in Fig 3a should have a log scale on the x-axis... would likely then look somewhat different. Isn't it surprising that they only see enrichment in each case (dots in the top right quadrant) – and never a de-enrichment of any markers (dots in the top left quadrant). Is that really correct?

There seems to be a cross-over between 'markers conferring increased resistance' and 'markers associated with functional T cell inhibition'.

Line 132 'Together, these results suggest that cells encoding intact proviruses and/or being part of large proviral clones display distinct phenotypic properties' - what do they think/propose is driving this?

156 Overall, inter-individually consistent surface marker expression differences between category 1 cells and HIV-negative cells were modest.

166 I did not understand this statement: By increasing the threshold for cellular activation, these markers may limit proviral gene expression and reduce subsequent visibility and vulnerability of reservoir cells to host immune recognition mechanisms.

Following 3 statements are all similar and important to their findings and general conclusions. I'm not entirely sure what they think has driven the changes – other than the obvious issue of immune selection – but how does this occur in the context of latent HIV infection – needs further explanation.

231 the upregulation of specific immunoregulatory markers on category 3 cells, including PD-L1, HVEM and PVR, was not selectively driven by specific cell clones but occurred relatively consistently across most analyzed clonal T cell populations

243 we found enrichment of reservoir cells with immune checkpoint molecules that control or restrict T cell activation, and likely reduce a cell's propensity to reactivate proviral gene transcription.

251 likely represent a consequence of immune selection mechanisms that promote preferential persistence of virally-infected cells with a phenotypic profile optimized towards minimizing exposure to and killing by host immune effector cells

Figure 4 is difficult to follow.

463 sites are listed when available. I didn't know how to follow these?

Referee #2 (Remarks to the Author):

Sun et al. used a novel single-cell, next-generation sequencing approach to characterize individual HIV-1-infected CD4 T cells from people living with HIV-1 receiving ART. They show that cells with defective proviruses are difficult to distinguish from their uninfected counterparts. In contrast, cells harbouring intact proviruses frequently expressed high levels of surface markers conferring resistance to immune-mediated killing as well as immune checkpoint molecules, which could contribute to their persistence. The authors conclude that only small subsets of infected cells with resistance to host immune effects are able to survive during long-term ART.

This is an interesting manuscript and the study is a technological tour-de-force as it allows the authors to determine the cell phenotype of cells carrying putatively intact HIV genomes (and their integration sites). The paper is well-written but more details should be provided when describing the figures (see minor comments). In addition, there are several limitations that dampen the enthusiasm for the study and that should be addressed.

1. My most important concern is the limited novelty of the findings. Although I commend the authors for developing this state-of-the-art single cell approach, the main conclusions of the study are largely confirmatory: It has been known for years that cells expressing immune checkpoint molecules are enriched in HIV reservoir and while some of these studies are quoted, others are not (Chew et al. Plos Pathogens 2016; Houry et al. JID 2017; Fromentin et al. Nat Comm 2019; Llewellyn et al. J Virol 2019 and many others). The field is already one step further with the evaluation of immune checkpoint blockers in cure studies (Lau et al. AIDS 2021; Uldrick et al. STM 2022; Baron et al. Cells 2022). Although the increased expression of ligands to negative receptors (HVEM, PVVR and PDL1) is new, this would require functional validation (see my third and fourth points).

2. The approach used by the authors to qualify « intact » proviruses is based on the sequencing of 18 regions of the viral genome. These 18 amplicons cover only a fraction of the full genome (maybe 60%?) and not its entirety, and proviral sequences from many cells are made of only a small number of amplicons (as seen in Supp Fig 2). In that sense, Figure 1C is misleading and should be modified to clearly show the gaps between these amplicons. Of course, a missing amplicon could be due to a real internal deletion or to an ineffective amplification and this complicates the analysis. The authors decided to use the 2 IPDA amplicons to determine the genetic intactness of these genomes. Since there are accumulating evidence that IPDA overestimates the reservoir (see Gaebler et al. JVI 2021 and Kinloch et al. Nat Comm 2021), it is unclear to which extent the approach used by the authors in here provides a correct assessment of the genetic intactness of the genomes.

3. All new markers identified as “enriching” in cells harbouring intact genomes should be experimentally validated by using conventional approaches: Cell sorting followed by 1) near full length genome sequencing and 2) viral outgrowth.

4. The author’s findings support the hypothesis that reservoir cells are selected during ART for their ability to escape immune killing. However, this is not directly demonstrated in the manuscript. The authors should show that reservoir cells are gradually enriched in subsets of cells expressing these markers over time (longitudinal analysis). They should also perform ex vivo killing assays to demonstrate that these reservoir cells persisting after prolonged therapy are relatively resistant to CTL and NK-mediated killing.

Minor points:

1. In the abstract, the authors used “markers associated with functional T cell inhibition” why not immune checkpoint molecules?
2. Category 4 does not appear in Supp Fig 1C.
3. Figure 1F should be better explained. Most panels in Figure 2 are not described in the main text.
4. The cell subsets (clusters) in Figure 2 should be better defined. Are TCM/TEM definitions based solely on CCR7 expression? What are Tmem#1 and #2? Why CD4 expression levels seem to differ between these subsets (Fig 2B).

Referee #3 (Remarks to the Author):

In this manuscript by Sun et al., the authors describe the application of a novel reservoir interrogation strategy, PheP-seq, that allows for phenotypic assessment of surface markers on HIV-infected cells at the single cell level. While other approaches have been previously published to phenotype HIV-infected cells, these have only been performed based on RNA or p24 expression, limiting the assay with a pre-stimulation step, an inability to assess the intactness of the provirus, and a likely biased dataset due to the inability to reactivate a pool of infected cells in deep latency. PheP-seq overcomes all of these caveats, representing a major step forward in the understanding of the ability to assess infected cells. Using PheP-seq, the authors describe the overall pools of HIV-infected cells, as well as cells harboring defective, hypermutated, and intact provirus, as well as cells representing clonally-expanded populations. In unbiased clustering, cells harboring intact or clonal provirus cluster mostly within central and effector memory populations, distinct from the overall pool of HIV-infected cells which do not differ in distribution from uninfected cells. Subsequently, they perform a formal statistical analysis of surface marker expression which reveals a number of markers, some novel and some previously identified, enriched on HIV-infected cells. Furthermore, they describe a unique profile of cells containing intact or clonal provirus, characterized by enrichment for inhibitory receptors and markers associated with protection from NK and T cell killing. The quality of the study is impressive and of great interest to the field. Below are my recommendations on how this manuscript could be improved.

Major Comments

1. The assay and analyses described in this manuscript represent significant progress in the ability to assess the phenotype of HIV-infected CD4 at a single cell resolution. However, the analysis is limited by its low throughput (193 cells with intact reservoir are assessed phenotypically) as well as its focus specifically on cells derived from peripheral blood. While acknowledging the difficulty of access to tissue samples from people living with HIV, the large majority of the reservoir is known to reside in lymphoid tissue and gut mucosa (Estes et al., *Nat Med*, 2017), sites which are likely to be phenotypically distinct. Therefore, while this manuscript does not suggest identification of a single marker that could serve as therapeutic targets for the HIV reservoir, the cellular description of intact cells in tissue may be distinct from that of blood, and is likely to be a better representation of the majority reservoir phenotype. If possible, the authors should confirm the most important data in cells from lymph node or gut mucosa. If not possible, this needs to be acknowledged and discuss.
2. It is difficult to understand the enrichment ratio in terms of biological significance, particularly given the low number of cells that seem to be analyzed for some markers. If a threshold has been chosen, the authors should clarify the minimum sensitivity value and enrichment values they consider for meaningful interpretation of the data. If a threshold has not been used, the authors should clarify how they have made subjective choices of which data are meaningful. As an example, TIGIT appears to be statistically significantly enriched in the Cat 1 vs Cat 0 comparison but only with an enrichment score of 1.16. Given that the authors imply that inhibitory receptors are selectively enriched on Cat 2 cells, they seem to not consider this a meaningful enrichment. Clarification on how these interpretations were made is critical.
3. While an impressive analysis, it is difficult to interpret the data in figure 5. The number of cells for each clone being analyzed should be annotated next to the clone identifier. If the “diversity” being seen is across a very small number of cells, probably it is not a sample size significant enough to

make conclusions from.

4. There appears to be discrepancies between the cluster distribution of intact cells in figure 2 and the marker enrichment described in figure 3. For example, in figure 2, cat 2 cells are almost entirely EM or CM, which are shown to be low in markers such as PD-1 and TIGIT, but in figure 3 there is an enrichment for these markers in cat 2 cells. The other markers that are discussed (PVR, HVEM, CD49d, CD45RO, CD95) do seem to match between the two figures. Are these enrichments being driven by the very few cat 2 cells in the Tmem #1 and Tmem #2 compartments? This same observation seems to also be true for cat 3 cells. It is unclear why the markers in Fig 2D were chosen to be displayed, rather than the markers that were identified later as being enriched (PVR, TIGIT, PD-1, HVEM etc.).

Referee #4 (Remarks to the Author):

The problem tackled here is very well motivated and the results interesting, but to be clear I am primarily reviewing this manuscript from a technical perspective. Truthfully, I found the paper to be a pleasure to read and technically seems to have been well done. As a non-HIV expert, I for the most part found it straightforward to follow, with some exceptions. Some comments, constructively offered:

1. PhEP-seq is effectively a means of concurrently profiling surface markers and transcriptome. I get that you are going for a specific set of sequences here, but it is still reminiscent of methods like CITE-seq. I'm wary of acronym proliferation, but deferring to the authors on that, some greater summary and hat-tipping of prior related methods (together with highlighting of differences) would be helpful.
2. I'm not an HIV expert, and I would have benefited from more handholding on the description of Category 1-4, e.g. something graphical re: what each means. Is it the case that all category 2 cells would have qualified for category 1, for example? Suggest adding a panel and rewriting paragraph on Page 4 to make more accessible to the non-expert. Fig 2C kind of gets at this but could be moved up and left me a little more confused (again, are category 2 cells also in category 1?)
3. Are these cells all from PBMCs? Would ideally mention in main text, along with any relevant points re: tissue resident CD4+ cells as a latent reservoir such that what you are seeing here is potentially a biased subset? Not my area of expertise but seems important to discuss.
4. I very much appreciate the measured approach to interpretation (e.g. some very significant results with modest fold changes may simply be due to the high numbers and not biologically significant). Good way to do science. Nonetheless, in places where you do thing the results are significant from an interpretive perspective, I would have like to have seen fold-changes in the text itself (e.g. page 6-7), i.e. more quantification along w/ your narrative.
5. Fig 1A is great, but I would have appreciated more detail in the main text (or if there is no room, maybe a Supplementary Note) about PhEP-seq, where it stands relative to other methods, narrative re: implementation by other groups to go along with methods, etc.
6. A point that I was wondering about after reading the discussion – is there any suggestion in the current data that profiling a much larger number of patients by this method would lead to 'structure' (e.g. clear subtypes of resistance/latency mechanisms), or is that really an unknown? I presume an

unknown b/c the n here is only 4 but I wonder if such an expanded study would be motivated to ask whether or not this is the case? Regardless, I think the conclusions in the last paragraph could be pulled back a little bit because although you maybe correct that there is not a universal footprint, it may well be the case that there is a finite number of subtypes/patterns that you can't really see with such a small n .

Author Rebuttals to Initial Comments:

Response to all Reviewers: We thank all reviewers for their very helpful comments and suggestions. As you can see from our responses, we have conducted extensive new experiments to address the issues raised; the new experimental data added to this manuscript include:

- Proteogenomic profiling of lymph node reservoir cells (n=396,628 total single cells, n=3,888 HIV-1 infected cells);
- Longitudinal proteogenomic profiling of reservoir cells from peripheral blood (n=257,574 total cells, n=4,156 HIV-1 infected cells);
- Orthogonal validation studies involving cell sorting and near full-length proviral sequencing, involving a total of n=852 single proviruses, of which n=95 were genome-intact;
- Functional assays to evaluate susceptibility/resistance of reservoir cells to CTL killing.

In total, we have analyzed proteogenomic profiles of 1,184,345 single memory CD4 T cells from both blood and lymph nodes, of which 10,903 are HIV-1-infected.

Many thanks in advance for re-reviewing this manuscript.

Referee #1:

The authors use novel single-cell, next-generation sequencing technology, which they combine with ex vivo phenotypic profiling of individual HIV-1-infected memory CD4 T cells to study the phenotype of latent HIV-infected cells.

This is one of the key questions in HIV biology and therefore of utmost importance and significance.

They find that cells harboring defective and therefore non-productive HIV-1 proviruses do not exhibit phenotypic differences compared to autologous uninfected cells. They found that phenotypic differences were more apparent for cells infected with HIV-1 genome-intact proviruses. Their overall conclusion seems to be that the changes in expression of cellular genes they observe are more likely to represent a consequence of immune selection i.e. these changes promote the preferential persistence of HIV-1 virally-infected cells with a phenotypic profile optimized towards minimizing exposure to and killing by host immune effector cells.

On the positive side their combination of single-cell, next-generation sequencing combined with ex vivo phenotyping of individual HIV-1-infected memory CD4 T cells describes the application of a nice technology for investigating the phenotype of latently- infected cells. Furthermore, they show differences between the cells harboring intact and incomplete HIV proviruses, and suggest some surface markers (Fig 3) and combinations of surface markers (Fig 4) which are differentially expressed. However, the differences seem to be very small, and (as they correctly state), the populations are obviously heterogeneous.

However, the overall interpretation is that the work is a "negative result" i.e. there is no really distinctive phenotype of latently infected cells with (or at least likely to have) intact HIV proviruses.

The question is therefore whether it's a sufficiently important negative result to justify publication in Nature - or whether, for instance, it's what everybody presumed to be the case anyway – which is perhaps more likely? Their results are obviously constrained by their choice of ~50 antibodies for cell phenotyping i.e. they could have missed interesting markers or

phenotypes by restricting themselves to this panel (compared with the 500 - 1000, or more, cell surface proteins present on the cells).

Many thanks for reviewing this work. Our study did not identify one specific surface marker that clearly distinguishes all HIV-infected cells from uninfected cells – we agree with the reviewer that the existence of such a marker is unlikely. However, we feel the true strength of our study is that it strongly supports the “immune selection hypothesis” of viral reservoir cell evolution: Our data indicate elevated expression of a specific set of phenotypic markers on HIV-1 reservoir cells that confer resistance to immune-mediated killing or reduce proviral transcriptional activity; enhanced expression of these markers results in a survival advantage, and suggests that only a small subset of reservoir cells with optimal adaptation to host immune responses can persist long-term.

In the revised version of the manuscript, we have extended these findings and now include phenotypic data for HIV-1 infected cells from lymph node samples. Interestingly, in lymph nodes, cells containing intact HIV-1 demonstrate a distinct phenotypic signature: In lymph nodes, HIV reservoir cells are most definitively distinguished by elevated expression of survival and anti-apoptosis markers (CD44, CD28, IL-21R, CD127); markers associated with resistance to immune killing were not as strongly increased on lymph node reservoir cells, possibly due to reduced cytotoxic activities of antiviral T cell responses in lymph nodes (Buggert et al, Cell 2020; Buggert, Plos Pathogens 2018; Nguyen et al, STM 2019), resulting in more limited selection pressure exerted by non-cytolytic lymph node-resident CD8 T cells. Collectively, our revised work suggests that HIV reservoir cells need to be optimally adapted to the immune microenvironment in their respective anatomical niche.

Points:

The manuscript might be made more accessible to the reader if the authors helped their audience: e.g. "...using a combination of X surface markers, we were able to identify Y% of the latently infected cells. Enrichment based on these markers suggests that 1/Z00 of these cells harbored intact proviruses..." (where 1/Z00 could be 1/100, 1/1,000 etc)...this would help the reader, and importantly they would then be able to validate their findings in independent experiments on different donors. Maybe the data has not been reported in this way, or validation experiment attempted, because it's just not possible i.e. no really "useful" surface markers were identified... (I note that the phenotypic signatures discussed are extremely general). They don't provide any orthogonal validation for their data...

In the revised version of the manuscript, we have now included sorting experiments for orthogonal validation of the markers for which HIV reservoir cells are enriched. These sorting experiments, performed with cells from peripheral blood (due to shortage of lymph node cells for sorting experiments), are consistent with the results observed with single-cell sequencing, although some inter-individual variations were noted. That said, we believe that these sorting experiments are less informative and provide a lot less granular and less quantitative information because sorting only classifies cells as being positive or negative for a given marker; as such, a continuous phenotyping variable obtained by single-cell next-generation sequencing is reduced to a categorical datapoint. A strength of our work is that cells are not simply being classified as positive or negative for a given phenotypic marker; instead, a continuous score for each phenotypic marker is collected by the PheP-Seq assay that more accurately reflects the phenotypic profile of HIV-1 reservoir cells. The sorting data are included in new **Supplemental Figures 6-7**.

Overall, it could (and as presented should) be interpreted as a "negative result" i.e. no really distinctive phenotype of latently infected cells with (or rather, likely to have) intact HIV proviruses.

It is true that our study did not identify one distinct marker that separates reservoir cells with intact proviruses from uninfected cells; instead, our study identified a combinatorial set of

phenotypic markers that distinguish HIV-infected from uninfected cells; this is true both in peripheral blood and lymph node cells. We propose that these markers are not constitutively elevated in viral reservoir cells, instead, their upregulation reflects a biological selection process during which only cells with elevated expression of these markers survive long-term. The identification of these markers contributes to the understanding of how HIV-1 reservoir cells resist immune clearance and persist for a lifetime; in addition, identifying these markers may offer opportunities for therapeutic targeting of viral reservoir cells in different tissue compartments.

To further support this concept, we have now also included longitudinal phenotyping data, showing that a distinct phenotypic signature of cells harboring intact proviruses is already present at earlier stages after ART initiation.

By strongly supporting the “immune selection hypothesis” of viral reservoir cell evolution, our study represents a distinct and novel finding – until recently, viral reservoir cells were considered as “stable” and “not susceptible to antiviral immune effects”. Our study strongly suggests that a considerable proportion of viral reservoir cells is susceptible to host immune effects, and that only a small subset of viral reservoir cells persists long term if their phenotypic properties promote survival in their specific immunological niche. Please note that our work also has important clinical implications: An intensification of immune activities (through e. g. therapeutic vaccines or other immunomodulatory interventions) may effectively eliminate viral reservoir cells that are optimally adapted to pre-existing autologous host immune responses.

Other points

They only analyse memory T cells (after magnetic enrichment). Some reports suggest that HIV proviruses may also be found in naive T cells (potentially accounting for a substantial minority of the total reservoir). It would therefore be interesting for them to discuss the pros and cons of this initial magnetic enrichment

We have considered the inclusion of naïve cells but due to technical considerations, this study was focused on memory cells, which harbor higher frequencies of viral reservoir cells. We propose that phenotypes of viral reservoir cells in the naïve CD4 T cell compartment could be considered in future dedicated studies.

-they seem to find intact HIV proviruses in cells with a mainly effector memory phenotype e.g. Fig 2a and text. Were they surprised by this as central memory T cells are implicated in the HIV reservoir - which makes sense, because they're the longest-lived T cells – this isn't discussed?

Thanks for raising this important point. We would like to point out that the classification of T cells in effector-memory and central-memory CD4 T cells is typically based on very few phenotypic markers (CD45RA and CCR7 in most cases); however, in our study, a much larger number of phenotypic parameters (>50) was analyzed, making the phenotypic classification of T cell subsets more complex. That said, it is true that a large proportion of HIV-1 reservoir cells appeared to be part of the effector-memory cell pool, although some reservoir cells with a central-memory cell/resident-memory pool were also detected. In particular, clonally-expanded reservoir cells seemed to be preferentially located in the effector-memory compartment; this is from our perspective consistent with elevated levels of clonal proliferation within the effector-memory compartment. It is possible that the central-memory cells represent precursor cells for the effector-memory cells, as has been suggested before.

-it would be useful to comment on the fraction of the HIV reservoir predicted to be found in peripheral blood cells, rather than secondary lymphoid tissue..... what implications does that have for their findings, which are all based on peripheral blood?

This is an excellent point. To address this, we now have conducted a proteogenomic analysis of HIV-1-infected cells from lymph nodes, as explained above. These studies show distinct phenotypic profiles of HIV-1-infected cells from blood vs lymph nodes, with weaker upregulation of negative immunoregulatory markers in lymph nodes, likely as a consequence of the predominance of non-cytolytic CD8 T cells in lymph nodes. Instead, lymph node reservoir cells demonstrated a profound upregulation of cell survival markers, including CD44, CD28, and the IL-21 receptor. Notably, in both blood and lymphoid tissues, the phenotypic properties of HIV-1-reservoir cells provide evidence for immune selection, and it appears that only reservoir cells with best adaptation to the local immune environment can persist long-term. New data on phenotypic profiles of HIV-1 reservoir cells in lymph nodes are shown in **Figure 5**.

Is it correct that the volcano plots in Fig 3a should have a log scale on the x-axis... would likely then look somewhat different. Isn't it surprising that they only see enrichment in each case (dots in the top right quadrant) – and never a de-enrichment of any markers (dots in the top left quadrant). Is that really correct?

We have chosen a linear scale for the x-axis for the volcano plots since the visual resolution is best with these settings. In the peripheral blood analysis, only a few markers were identified for which a small, non-significant trend for de-enrichment in HIV reservoir cells was noted. In lymph node cells, markers with de-enrichment for specific markers among viral reservoir cells were more obvious. We would like to highlight that most of the markers included in our analysis were pre-selected based on the hypothesis that they may be upregulated on HIV-1 reservoir cells; this may also in part explain why we did not observe markers that were downregulated on HIV-1-infected cells from peripheral blood.

There seems to be a cross-over between 'markers conferring increased resistance' and 'markers associated with functional T cell inhibition'.

It is true that these markers are biologically related, however, there is no direct overlap. In some cases, markers that confer resistance to immune-mediated killing act as receptors for markers associated with functional T cell inhibition (e. g. PD-L1 (resistance marker) is the receptor for PD-1 (functional inhibition marker).

Line 132 'Together, these results suggest that cells encoding intact proviruses and/or being part of large proviral clones display distinct phenotypic properties' - what do they think/propose is driving this?

We propose that immune-mediated selection mechanisms are responsible for this: Cells that encode for intact HIV-1 and large clones (which presumably have a higher chance to reactivate proviral gene expression during proliferation) are more readily recognized by immune cells and for this reason are under more intense immune selection pressure. Due to this immune selection process, only cells with specific phenotypic properties can persist long-term; alternative cells without such phenotypic markers were likely successfully eliminated.

156 Overall, inter-individually consistent surface marker expression differences between category 1 cells and HIV-negative cells were modest.

To address interindividual differences, we have used a bootstrapping analysis approach so that each patient is equally considered in the statistical evaluation. We agree that the enrichment for individual markers in category 1 cells is frequently modest, possibly due to the fact that many defective proviruses are only producing limited amounts of viral RNA or proteins and are under more limited immune selection pressure. However, distinct phenotypic signs were more definitive on category 2 viral reservoir cells in both blood and lymph nodes.

166 I did not understand this statement: By increasing the threshold for cellular activation, these markers may limit proviral gene expression and reduce subsequent visibility and vulnerability of reservoir cells to host immune recognition mechanisms.

This statement refers to markers associated with functional inhibition of T cells. It is well recognized that infected T cells may reactivate or increase viral transcription upon cellular activation. The functional inhibition markers reduce T cell activation and through this mechanism may block proviral transcriptional activity. This perspective is supported by functional studies demonstrating increased proviral transcription of HIV following experimental blockade of these markers, as discussed in the manuscript.

Following 3 statements are all similar and important to their findings and general conclusions. I'm not entirely sure what they think has driven the changes – other than the obvious issue of immune selection – but how does this occur in the context of latent HIV infection – needs further explanation.

231 the upregulation of specific immunoregulatory markers on category 3 cells, including PD-L1, HVEM and PVR, was not selectively driven by specific cell clones but occurred relatively consistently across most analyzed clonal T cell populations

243 we found enrichment of reservoir cells with immune checkpoint molecules that control or restrict T cell activation, and likely reduce a cell's propensity to reactivate proviral gene transcription.

251 likely represent a consequence of immune selection mechanisms that promote preferential persistence of virally-infected cells with a phenotypic profile optimized towards minimizing exposure to and killing by host immune effector cells

We propose that HIV reservoir cells, specifically those that encode for intact HIV-1 proviruses or are clonally-expanded, are under continuous immune selection pressure. This “immune selection hypothesis” is based on recent experimental advances showing that a considerable proportion of HIV-1 reservoir cells remains transcriptionally active during suppressive antiretroviral therapy and can be visible to immune recognition mechanisms (Einkauf et al, Cell 2022). Please note that immune selection of HIV-1 reservoir cells is also supported by recent studies using proviral chromosomal integration sites as biomarkers of immune selection processes (Seiger et al, CROI 2022, abstract 68).

Figure 4 is difficult to follow.

We apologize for the complexity of this figure; we have improved the graphics of the figure to make it more accessible.

463 sites are listed when available. I didn't know how to follow these?

We have edited and re-designed the figure and figure legend for Figure 2 to address this.

Referee #2:

Sun et al. used a novel single-cell, next-generation sequencing approach to characterize individual HIV-1-infected CD4 T cells from people living with HIV-1 receiving ART. They show that cells with defective proviruses are difficult to distinguish from their uninfected counterparts. In contrast, cells harbouring intact proviruses frequently expressed high levels of surface markers conferring resistance to immune-mediated killing as well as immune checkpoint molecules, which could contribute to their persistence. The authors conclude that only small subsets of infected cells with resistance to host immune effects are able to survive during long-term ART.

This is an interesting manuscript and the study is a technological tour-de-force as it allows the authors to determine the cell phenotype of cells carrying putatively intact HIV genomes (and their integration sites). The paper is well-written but more details should be provided when describing the figures (see minor comments). In addition, there are several limitations that dampen the enthusiasm for the study and that should be addressed.

1. My most important concern is the limited novelty of the findings. Although I commend the authors for developing this state-of-the-art single cell approach, the main conclusions of the study are largely confirmatory: It has been known for years that cells expressing immune checkpoint molecules are enriched in HIV reservoir and while some of these studies are quoted, others are not (Chew et al. Plos Pathogens 2016; Khoury et al. JID 2017; Fromentin et al. Nat Comm 2019; Llewellyn et al. J Virol 2019 and many others). The field is already one step further with the evaluation of immune checkpoint blockers in cure studies (Lau et al. AIDS 2021; Uldrick et al. STM 2022; Baron et al. Cells 2022). Although the increased expression of ligands to negative receptors (HVEM, PVVR and PDL1) is new, this would require functional validation (see my third and fourth points).

There are a number of key issues in which our data differ from previous studies:

- In our work, the phenotype of viral reservoir cells was directly assessed *ex vivo*, and not after *in vitro* stimulation like in previous studies; in fact, previous studies were unable to evaluate whether cells that express higher levels of specific phenotypic markers after *in vitro* stimulation do so due to higher responsiveness to *in vitro* reactivation stimuli or due to a higher expression of these markers at baseline;
- Prior studies, including the ones cited by the reviewer, did not distinguish between cells harboring intact proviruses or defective proviruses. This distinction, however, is critical based on our results;
- Prior results only evaluated the small subset of reservoir cells that were responsive to *in vitro* reactivation; the phenotype of the larger population of cells unable to respond to reactivation remained entirely unclear;
- A much larger number of phenotypic markers ($n > 50$) is evaluated in our study relative to previous work. The list of surface markers that are differentially expressed on reservoir cells in our study includes a diverse list of markers, many of which were not evaluated at all in previous work;
- The differentially-expressed phenotypic markers on HIV-1 reservoir cells also predominantly include markers that have not been associated with phenotypic signatures of HIV-1 reservoir cell biology in the past but show remarkable upregulation on HIV-1-infected cells encoding for intact HIV-1. This, for example, is true for TGF- β -Receptor expression, IL-21 Receptor expression, KLRG-1, BTLA, HLA-E, PVR, HVEM, CD127;
- One single marker, such as PD-1, only defines a relatively small proportion of reservoir cells in a given ART-treated individual, however, as shown in Figure 3, the proportions of reservoir cells harboring intact proviruses are significantly higher when a combinatorial set of markers is considered. This finding will be relevant for future clinical cure studies with combined immunotherapeutic interventions;
- Please also note that our study represents the first one to assess *ex vivo* phenotypic features of viral reservoir cells (encoding for intact or defective proviruses) on a clonal level; this is achieved through simultaneous amplification of HIV-1 DNA and corresponding (pre-identified) viral-host junctions.

In addition, we have addressed this most important concern of reviewer 2 by adding a new dataset describing phenotypic properties of HIV-1 reservoir cells from three different lymph nodes from ART-treated persons. These experiments demonstrated that cells encoding for intact HIV-1 in lymph nodes from persons after approx. 10 years of ART are characterized by distinct surface expression of CD28, CD127, CD44 and the IL-21R, all of which contribute to cell survival. Markers associated with resistance to immune-mediated killing were less profoundly upregulated in lymph node reservoir cells, possibly because CD8 T cells from LN

have lower cytolytic activities and exert less immune selection pressure in this specific tissue compartment (Buggert et al, Cell 2020; Buggert, Plos Pathogens 2018; Nguyen et al, STM 2019). These novel observations suggest a compartmentalized immune adaptation of HIV-1 reservoir cells to their specific microanatomical immune environment.

2. The approach used by the authors to qualify « intact » proviruses is based on the sequencing of 18 regions of the viral genome. These 18 amplicons cover only a fraction of the full genome (maybe 60%?) and not its entirety, and proviral sequences from many cells are made of only a small number of amplicons (as seen in Supp Fig 2). In that sense, Figure 1C is misleading and should be modified to clearly show the gaps between these amplicons. Of course, a missing amplicon could be due to a real internal deletion or to an ineffective amplification and this complicates the analysis. The authors decided to use the 2 IPDA amplicons to determine the genetic intactness of these genomes. Since there are accumulating evidence that IPDA overestimates the reservoir (see Gaebler et al. JVI 2021 and Kinloch et al. Nat Comm 2021), it is unclear to which extent the approach used by the authors in here provides a correct assessment of the genetic intactness of the genomes.

The IPDA assay, published in *Nature* in 2019 and since then cited more than 300 times, has become a widely accepted method for analyzing intact proviruses, and is frequently used as a benchmark across the entire HIV research community. This is likely due to the extensive experimental work up that the Siliciano lab conducted to evaluate and validate this assay. When they occur, IPDA problems are typically related to sequence mismatches in primer binding regions that are due to HIV-1 polymorphisms, as described by Kinloch et al. However, this is not a concern in our study, since full-genome sequencing data for intact proviruses were known upfront for the study subjects, and primers were adjusted and customized to the autologous patient-derived sequences. Please note that the small amplicon size of the IPDA can be viewed as an advantage of the IPDA, since the PCR efficacy for such small amplicons is very high (Gaebler et al, JVI 2021).

In addition, our technology allows us to unequivocally identify intact proviruses based on their chromosomal integration sites; this is enabled by single-cell amplification of viral-host junctions for proviruses for which genome-intactness has been previously determined using near full-genome sequencing. The phenotypes of cells that harbor intact proviruses determined by this integration site-based approach do not differ from cells harboring intact proviruses determined by IPDA, again supporting the important work from the Siliciano lab showing that IPDA is a reliable approach for identifying intact proviruses.

Figure 1C is a continuation of Figure 1B, in which 18 HIV amplicons and the gaps between amplicons are indicated. We have adjusted the figure legend to emphasize this point. Figure 1C is meant to zoom in to show the detection of each amplicon from each cell; therefore, we display all 18 amplicons together.

3. All new markers identified as “enriching” in cells harbouring intact genomes should be experimentally validated by using conventional approaches: Cell sorting followed by 1) near full length genome sequencing and 2) viral outgrowth.

We have conducted sorting experiments with peripheral blood cells to address this point, using a full-genome sequencing approach. The data collected in these experiments, involving an assessment of 852 individual proviruses, among which 95 were genome-intact, support our prior conclusions (**new supplemental Figures S6-7**), although some inter-individual variations were noted. Based on our analysis, we propose that biological differences are not conferred by a single isolated marker; instead, it is likely that combinations of markers are determining a cell’s biological behavior; for this reason, we have performed sorting experiments with combinations of markers. Moreover, it is technically not possible to individually sort cells positive for each differentially-expressed marker; due to the relatively

high number of markers that were used and the frequent overlap between cells expressing these markers, this would be impossible with a 4-population sorting device (BD Aria). We would also like to point out that we consider such sorting experiments as technically less informative relative to the single-cell proteogenomic profiling approaches described in our work. Most importantly, sorting does not reveal the direct expression intensity of a given marker, but instead relies on transforming continuous phenotypic profiling data into categorical data (+ or -). Please also note that the single-cell proteogenomic profiling approach used in our experiments has already been extensively validated during its design and implementation process (see Ruff et al, *Methods Mol Biol* 2022 (PMID: 34766272)); data from this platform are, for example, used for therapeutic/clinical decision-making in the context of hematologic malignancies, and have been presented in the context of hematologic cancers in multiple prior publications (e. g. Miles et al, *Nature* 2020 (PMC7677169); Demaree et al, *Nat Com* 2021 (PMC7952600)).

We ask for the reviewer's understanding that we were unable to conduct *in vitro* viral outgrowth assays with sorted subpopulations of cells. Such experiments require large volumes of cells (obtained from leukapheresis), since *in vitro* viral outgrowth experiments only capture a very small proportion of all intact proviruses, typically around or less than 5% of the genome-intact proviruses detected by near full-length sequencing. Unfortunately, all leukapheresis cell collection protocols for HIV research purposes were on hold at our institution during the covid pandemic (due to prioritization of leukapheresis protocols for covid research), and leukapheresis samples from HIV patients collected prior to the covid pandemic were depleted. We would also like to point out that *in vitro* viral outgrowth is profoundly influenced by stochastic noise (well summarized in e. g. Hansen et al, *HIV Latency: Stochastic across Multiple Scales, Cell Host & Microbe* 2019), and a 21-day viral outgrowth assay (the protocol used in the vast majority of cases) is unreliable for detecting the true number of replication-competent proviruses (Ho et al, *Cell* 2014). In fact, Hataye et al. (*Cell Host & Microbe*, 2019 (PMC6948011)) have shown that cells in which proviral latency is effectively reversed in *in vitro* outgrowth assays are more likely to die during the subsequent 21-day culture than to initiate exponential viral growth, contributing to the stochastic nature of results from viral outgrowth assays. For these reasons, we have conducted near full-length proviral sequencing as a more quantitative assessment of intact proviruses in the sorted cell subpopulations.

4. The author's findings support the hypothesis that reservoir cells are selected during ART for their ability to escape immune killing. However, this is not directly demonstrated in the manuscript. The authors should show that reservoir cells are gradually enriched in subsets of cells expressing these markers over time (longitudinal analysis). They should also perform *ex vivo* killing assays to demonstrate that these reservoir cells persisting after prolonged therapy are relatively resistant to CTL and NK-mediated killing.

We have performed single-cell proteogenomic profiling on longitudinal samples from two patients for whom samples from earlier time points were available. These results, shown in **Figure 3B**, demonstrate that the distinct phenotypic profile of cells harboring intact HIV-1 proviruses is quite stable over almost a decade and already detectable at early stages after ART initiation. This suggests that reservoir cells encoding for intact HIV-1 are likely to have adapted very quickly to their immune environment and thus have evolutionarily advantageous phenotypic properties from the early stages of antiretroviral treatment. This is consistent with the proposed immune selection hypothesis; however, it appears that immune selection of reservoir cells may already occur at the early stage of ART, and possibly even earlier than that. Future studies using samples collected weeks and months after ART initiation, and prior to ART initiation, will be helpful to further dissect the evolutionary dynamics of viral reservoir cells.

In addition, we have performed functional assays to evaluate the susceptibility/resistance of viral reservoir cells to immune-mediated killing. To preserve the original physiological

phenotype of the reservoir cells and minimize artifacts from in vitro culture, we have subjected ex-vivo isolated reservoir cells to HIV-1-specific T cell clones, in the presence or absence of cognate peptide antigens; following co-culture with clones, the number of intact and defective proviruses were determined using the IPDA assay. We observed that the relative frequency of intact proviruses in patient-derived cells is increased after exposure to CTL clones, indicating that cells infected with intact proviruses were more likely to persist and resist T cell-mediated killing compared to HIV-uninfected cells.

The results support the hypothesis that viral reservoir cells, after 10 years of continuous ART, have increased resistance to immune-mediated killing, likely as a result of immune selection mechanisms leading to a survival advantage of cells with lower susceptibility to CTL. These new results are shown in **Supplemental Figure 8**. Please note that functional assays showing resistance to immune-mediated killing of reservoir cells have also been described in the important work conducted by Dr. Brad Jones, published in two manuscripts in the *Journal of Clinical Investigation* (refs 39,40 in the manuscript).

Minor points:

1. In the abstract, the authors used “markers associated with functional T cell inhibition” why not immune checkpoint molecules?

We have corrected this and now use both terms interchangeably in our manuscript.

2. Category 4 does not appear in Supp Fig 1C.

We have added these numbers to the manuscript.

3. Figure 1F should be better explained. Most panels in Figure 2 are not described in the main text.

We have edited and re-designed the figures to address this.

4. The cell subsets (clusters) in Figure 2 should be better defined. Are TCM/TEM definitions based solely on CCR7 expression? What are Tmem#1 and #2? Why CD4 expression levels seem to differ between these subsets (Fig 2B).

Yes, TCM/TEM definitions were defined based on CCR7. Unfortunately, the Tmem#1 and #2 cells cannot be more specifically characterized in such a global analysis involving more than 50 phenotypic markers; this is a notable difference to flow cytometry where the classification of T cell subsets involves a lot fewer markers. The global UMAP plots are mostly used to demonstrate the positioning of HIV-1-infected cells relative to total memory CD4 T cells. The heatmap in Figure 2B shows expression intensities of markers normalized among all T cell subsets. It is true that CD4 expression varies among these memory CD4 T cell subpopulations. As expected, CD4 expression seems downregulated on activated CD4 T cell subsets.

Referee #3:

In this manuscript by Sun et al., the authors describe the application of a novel reservoir interrogation strategy, PheP-seq, that allows for phenotypic assessment of surface markers on HIV-infected cells at the single cell level. While other approaches have been previously published to phenotype HIV-infected cells, these have only been performed based on RNA or p24 expression, limiting the assay with a pre-stimulation step, an inability to assess the intactness of the provirus, and a likely biased dataset due to the inability to reactivate a pool of infected cells in deep latency. PheP-seq overcomes all of these caveats, representing a

major step forward in the understanding of the ability to assess infected cells. Using PheP-seq, the authors describe the overall pools of HIV-infected cells, as well as cells harboring defective, hypermutated, and intact provirus, as well as cells representing clonally-expanded populations. In unbiased clustering, cells harboring intact or clonal provirus cluster mostly within central and effector memory populations, distinct from the overall pool of HIV-infected cells which do not differ in distribution from uninfected cells. Subsequently, they perform a formal statistical analysis of surface marker expression which reveals a number of markers, some novel and some previously identified, enriched on HIV-infected cells. Furthermore, they describe a unique profile of cells containing intact or clonal provirus, characterized by enrichment for inhibitory receptors and markers associated with protection from NK and T cell killing. The quality of the study is impressive and of great interest to the field. Below are my recommendations on how this manuscript could be improved.

Major Comments

1. The assay and analyses described in this manuscript represent significant progress in the ability to assess the phenotype of HIV-infected CD4 at a single cell resolution. However, the analysis is limited by its low throughput (193 cells with intact reservoir are assessed phenotypically) as well as its focus specifically on cells derived from peripheral blood. While acknowledging the difficulty of access to tissue samples from people living with HIV, the large majority of the reservoir is known to reside in lymphoid tissue and gut mucosa (Estes et al., Nat Med, 2017), sites which are likely to be phenotypically distinct. Therefore, while this manuscript does not suggest identification of a single marker that could serve as therapeutic targets for the HIV reservoir, the cellular description of intact cells in tissue may be distinct from that of blood, and is likely to be a better representation of the majority reservoir phenotype. If possible, the authors should confirm the most important data in cells from lymph node or gut mucosa. If not possible, this needs to be acknowledged and discuss.

We have conducted a phenotypic analysis of HIV-1 reservoir cells from inguinal lymph nodes in three ART-treated individuals. As the reviewer seems to have anticipated, the phenotype of such HIV-1 reservoir cells in lymph nodes shows some distinct features relative to peripheral blood cells. In particular, enrichment of reservoir cells with markers associated with resistance to immune-mediated killing seems more limited in lymph nodes, likely due to an altered immune microenvironment in lymph nodes that does not include highly cytolytic CTL (Buggert et al, Cell 2020; Nguyen et al, STM 2019); instead, reservoir cells from lymph nodes show increased expression of markers associated with cell survival (e. g. CD127, CD44, IL-21R, CD28). As also outlined in our response to reviewer 1, we suggest that HIV-1 reservoir cell phenotypes display a compartmentalized adaptation to the specific immunological niche they are residing in.

2. It is difficult to understand the enrichment ratio in terms of biological significance, particularly given the low number of cells that seem to be analyzed for some markers. If a threshold has been chosen, the authors should clarify the minimum sensitivity value and enrichment values they consider for meaningful interpretation of the data. If a threshold has not been used, the authors should clarify how they have made subjective choices of which data are meaningful. As an example, TIGIT appears to be statistically significantly enriched in the Cat 1 vs Cat 0 comparison but only with an enrichment score of 1.16. Given that the authors imply that inhibitory receptors are selectively enriched on Cat 2 cells, they seem to not consider this a meaningful enrichment. Clarification on how these interpretations were made is critical.

In single-cell experiments, large numbers of individual datapoints are collected, therefore, significance levels, even when adjusted for multiple testing and corrected by a bootstrapping approach (so that each patient's data make a similar statistical contribution to the final data analysis) can be very high. Therefore, we think that biological judgment is necessary to interpret these findings. In our case, we considered three parameters to identify markers that

are likely to be biologically meaningfully altered on HIV-1 reservoir cells: the fold-change in expression intensity (relative to reference cell subsets, typically >1.5 fold), the associated FDR-adjusted p-value (<0.05) and the relative proportion of cells expressing a given marker (typically >25%).

3. While an impressive analysis, it is difficult to interpret the data in figure 5. The number of cells for each clone being analyzed should be annotated next to the clone identifier. If the “diversity” being seen is across a very small number of cells, probably it is not a sample size significant enough to make conclusions from.

We have followed the reviewer’s advice; the numbers of cells are now listed in the updated **Figure 4** (previous figure 5). The data displayed were from large clones with more than 5 clonally-expanded reservoir cells identified in the analysis.

4. There appears to be discrepancies between the cluster distribution of intact cells in figure 2 and the marker enrichment described in figure 3. For example, in figure 2, cat 2 cells are almost entirely EM or CM, which are shown to be low in markers such as PD-1 and TIGIT, but in figure 3 there is an enrichment for these markers in cat 2 cells. The other markers that are discussed (PVR, HVEM, CD49d, CD45RO, CD95) do seem to match between the two figures. Are these enrichments being driven by the very few cat 2 cells in the Tmem #1 and Tmem #2 compartments? This same observation seems to also be true for cat 3 cells. It is unclear why the markers in Fig 2D were chosen to be displayed, rather than the markers that were identified later as being enriched (PVR, TIGIT, PD-1, HVEM etc.).

The reason for these observations is that in the UMAP plots on Figure 2, marker expression is shown across (and normalized within) the total pool of memory CD4 T cells; since reservoir cells only represent a very small component of the total memory CD4 T cells, the individual expression of surface markers on HIV-1 reservoir cells is not well reflected in the UMAP plot – the UMAP plots are only shown for a global analysis of reservoir cells relative to the total pool of all memory CD4 T cells. In Figure 2C, the phenotypic profile of HIV-1 reservoir cells is directly contrasted to the HIV-1-uninfected cells in this type of analysis.

Referee #4:

The problem tackled here is very well motivated and the results interesting, but to be clear I am primarily reviewing this manuscript from a technical perspective. Truthfully, I found the paper to be a pleasure to read and technically seems to have been well done. As a non-HIV expert, I for the most part found it straightforward to follow, with some exceptions. Some comments, constructively offered:

1. PhEP-seq is effectively a means of concurrently profiling surface markers and transcriptome. I get that you are going for a specific set of sequences here, but it is still reminiscent of methods like CITE-seq. I’m wary of acronym proliferation, but deferring to the authors on that, some greater summary and hat-tipping of prior related methods (together with highlighting of differences) would be helpful.

Many thanks for your comments and for reviewing this paper from a technical perspective. The main difference between our approach and the CITE-Seq is that we analyze HIV-1 DNA integrated into the host genome in conjunction with phenotypic markers; in CITE-Seq, phenotypic markers are analyzed with RNA expression patterns. We would be happy to elaborate on this in the final version of the manuscript if space permits.

2. I’m not an HIV expert, and I would have benefited from more handholding on the description of Category 1-4, e.g. something graphical re: what each means. Is it the case that all category 2 cells would have qualified for category 1, for example? Suggest adding a panel and rewriting

paragraph on Page 4 to make more accessible to the non-expert. Fig 2C kind of gets at this but could be moved up and left me a little more confused (again, are category 2 cells also in category 1?)

We have revised the manuscript to clarify this point: Category I cell include all cells that contain HIV-1 DNA of any kind; category II, category III and category IV cells represent subgroups of category I.

3. Are these cells all from PBMCs? Would ideally mention in main text, along with any relevant points re: tissue resident CD4+ cells as a latent reservoir such that what you are seeing here is potentially a biased subset? Not my area of expertise but seems important to discuss.

Thanks to the reviewer for pointing this out. We have now included a large new dataset that specifically describes the phenotypic profile of HIV-1 reservoir cells from lymph nodes to address this point.

4. I very much appreciate the measured approach to interpretation (e.g. some very significant results with modest fold changes may simply be due to the high numbers and not biologically significant). Good way to do science. Nonetheless, in places where you do thing the results are significant from an interpretive perspective, I would have like to have seen fold-changes in the text itself (e.g. page 6-7), i.e. more quantification along w/ your narrative.

Many thanks for this comment. Our main concern is that the manuscript is already quite long, and will likely have to be further shortened should it move to publication. For this reason, we don't spell out the fold-changes but instead have included them in the figures and in our supplemental Tables.

5. Fig 1A is great, but I would have appreciated more detail in the main text (or if there is no room, maybe a Supplementary Note) about PhEP-seq, where it stands relative to other methods, narrative re: implementation by other groups to go along with methods, etc.

Again, this is a matter of available space, which is unfortunately limited. If this manuscript proceeds, we will do our best to include edits to address this point.

6. A point that I was wondering about after reading the discussion – is there any suggestion in the current data that profiling a much larger number of patients by this method would lead to 'structure' (e.g. clear subtypes of resistance/latency mechanisms), or is that really an unknown? I presume an unknown b/c the n here is only 4 but I wonder if such an expanded study would be motivated to ask whether or not this is the case? Regardless, I think the conclusions in the last paragraph could be pulled back a little bit because although you maybe correct that there is not a universal footprint, it may well be the case that there is a finite number of subtypes/patterns that you can't really see with such a small n.

Many thanks for this insightful comment – we have adjusted the discussion to address this point.

Reviewer Reports on the First Revision:

Referees' comments:

Referee #1 (Remarks to the Author):

Many thanks for asking me to review the rebuttal to this paper. The authors have made considerable and important additions to the initial submission.

In particular they have now profiled a significant number of lymph node reservoir cells as well as longitudinal profiling of reservoir cells and some orthogonal validation studies. These data have considerably strengthened the manuscript and while there is no 'magic bullet' cell surface marker identified – the findings are both important for the field and robust. The presentation has also improved significantly.

Referee #2 (Remarks to the Author):

I appreciate the author's response to my comments. The authors have adequately addressed most of my concerns. I have a few additional suggestions:

1. The addition of the phenotypic analysis of infected CD4+ T cells in lymph node is remarkable and the observation that cellular features contributing to the persistence of HIV-infected cells differ between blood and lymphoid tissues is fascinating. I am not sure why CD127 (IL-7 receptor), which is definitely associated with T cell survival and one of the top receptors expressed by cells with intact genomes was not mentioned in the abstract.
2. I also appreciate the effort made by the authors to validate their findings by sorting discrete populations of cells based on the markers they identified as enriched for intact HIV genomes using their sophisticated approach. The results presented Supp Figures 6 and 7 demonstrate clear enrichment for intact HIV DNA in cells expressing some of these receptors in the majority of the 3 to 4 samples tested. However, it is unclear how the gates were set in these experiments : it looks like memory CD4+ T cells do not express HVEM, PVR, PDL1 and HLA-E. Please explain.
3. I understand and acknowledge the limitations of the viral outgrowth assay and agree with the author's response on that point.
4. The new longitudinal study (Figure 3B) is also interesting but does not support the concept of a selection process during prolonged ART, since there was no clear change in the phenotypic signature of cells with intact genomes over years of therapy. My understanding is that the phenotype of cells with intact genomes is similar after short and long term ART. This would suggest that rather than a selection process over years of ART, this phenomenon occurs very quickly upon ART initiation. The authors may want to convey this message more clearly.
5. Figure 1F is still not explained in the main text and should probably be moved to supplementary.
6. Line 207 : "Cells belonging to the same clone typically showed relatively little variation and tended to cluster near one another on a UMAP plot, suggesting a rather homogenous phenotypic behavior of HIV-1-infected cells derived from the same clone (Figure 4A)". I respectfully disagree with this

conclusion. Figure 4A clearly shows that these clones do not form unique clusters and belong to different subsets (as defined by the authors in Figure 2). I suggest to modify Figure 4A to keep the color code of different clusters as in Figure 2, and to modify the above sentence.

Referee #3 (Remarks to the Author):

The authors addressed all of mine major points, including the complex one of adding data for lymph node cells, and did a large amount of additional work to address other reviewers. The new data on LN infected cells, with a phenotype distinct from those in blood, are very interesting and consistent with recent work showing poor cytolytic responses in LN.

Referee #4 (Remarks to the Author):

The paper is considerably stronger, and my comments, which were relatively few, were generally addressed. One point that was not addressed is that there remains no/minimal discussion of this newly named method, Phep-seq, in relation to other methods in the literature. The authors point to cite limitations, but there is no reason why they couldn't include a brief supplementary note or add one sentence with citations to other papers that have done concurrent cellular marker phenotyping currently with nucleic acid analysis (e.g. CITE-seq, DAb-seq [which is immunophenotyping + DNA; see Demaree et al. Nature Communications 2021], etc.). The authors are correct that the method is different from CITE-seq, but there are similarities; DAb-seq is also relevant as an earlier method implementing a DNA/protein co-assay at single cell resolution. I recognize the unique value of what's been developed here for HIV in particular, but this is a pretty minimal ask and a failure to cite or mention the substantial work that's been done in this area is misleading.

Author Rebuttals to First Revision:

Response to Reviewers

Referee #1:

Many thanks for asking me to review the rebuttal to this paper. The authors have made considerable and important additions to the initial submission.

In particular they have now profiled a significant number of lymph node reservoir cells as well as longitudinal profiling of reservoir cells and some orthogonal validation studies. These data have considerably strengthened the manuscript and while there is no 'magic bullet' cell surface marker identified – the findings are both important for the field and robust. The presentation has also improved significantly.

Many thanks to reviewer 1 for reviewing this paper.

Referee #2:

I appreciate the author's response to my comments. The authors have adequately addressed most of my concerns. I have a few additional suggestions:

1. The addition of the phenotypic analysis of infected CD4+ T cells in lymph node is remarkable and the observation that cellular features contributing to the persistence of HIV-infected cells differ between blood and lymphoid tissues is fascinating. I am not sure why CD127 (IL-7 receptor), which is definitely associated with T cell survival and one of the top receptors expressed by cells with intact genomes was not mentioned in the abstract.

We have now mentioned CD127 in the abstract.

2. I also appreciate the effort made by the authors to validate their findings by sorting discrete populations of cells based on the markers they identified as enriched for intact HIV genomes using their sophisticated approach. The results presented Supp Figures 6 and 7 demonstrate clear enrichment for intact HIV DNA in cells expressing some of these receptors in the majority of the 3 to 4 samples tested. However, it is unclear how the gates were set in these experiments: it looks like memory CD4+ T cells do not express HVEM, PVR, PDL1 and HLA-E. Please explain.

We have included flow cytometry panels to demonstrate our gating process in the Extended Data Figures; these data are showing the surface expression of these markers on memory CD4 T cells and, for comparative purposes, on non-CD4 cells. The gates have been set according to FMO controls. It is true that the expression intensity of HVEM, PVR, PDL1 and HLA-E is relatively limited on the total memory CD4 T cells, however, our sorting experiments demonstrated that cell populations that do express these markers are enriched for cells encoding for intact HIV-1.

3. I understand and acknowledge the limitations of the viral outgrowth assay and agree with the author's response on that point.

Many thanks.

4. The new longitudinal study (Figure 3B) is also interesting but does not support the concept of a selection process during prolonged ART, since there was no clear change in the phenotypic signature of cells with intact genomes over years of therapy. My understanding is that the phenotype of cells with intact genomes is similar after short and long term ART. This would suggest that rather than a selection process over years of ART, this phenomenon occurs very quickly upon ART initiation. The authors may want to convey this message more clearly.

We have modified the manuscript text to address this. For some markers (HVEM, CD127, PVR, TGF- β R), we do observe an increase in surface expression on cells during prolonged ART; for other markers, there is no notable difference between cells collected at early vs. later timepoints after ART initiation; we, therefore, propose that selection may occur both early after ART initiation, as well as later during prolonged ART.

5. Figure 1F is still not explained in the main text and should probably be moved to supplementary.

We have followed the reviewer's advice and moved this figure into the extended data.

6. Line 207: "Cells belonging to the same clone typically showed relatively little variation and tended to cluster near one another on a UMAP plot, suggesting a rather homogenous phenotypic behavior of HIV-1-infected cells derived from the same clone (Figure 4A)". I respectfully disagree with this conclusion. Figure 4A clearly shows that these clones do not form unique clusters and belong to different subsets (as defined by the authors in Figure 2). I suggest to modify Figure 4A to keep the color code of different clusters as in Figure 2, and to modify the above sentence.

Many thanks for this comment; we have changed the language as suggested by the reviewer. While clones in many cases cluster together on a UMAP plot, they sometimes display phenotypic diversity and individual clone members can belong to different computationally-defined memory cell subsets.

Referee #3:

The authors addressed all of mine major points, including the complex one of adding data for lymph node cells, and did a large amount of additional work to address other reviewers. The new data on LN infected cells, with a phenotype distinct from those in blood, are very interesting and consistent with recent work showing poor cytolytic responses in LN.

Thanks for your comments.

Referee #4:

The paper is considerably stronger, and my comments, which were relatively few, were generally addressed. One point that was not addressed is that there remains no/minimal discussion of this newly named method, Phep-seq, in relation to other methods in the literature. The authors point to cite limitations, but there is no reason why they couldn't include a brief supplementary note or add one sentence with citations to other papers that have done concurrent cellular marker phenotyping currently with nucleic acid analysis (e.g. CITE-seq, DAb-seq [which is immunophenotyping + DNA; see Demaree et al. Nature Communications 2021], etc.). The authors are correct that the method is different from CITE-seq, but there

are similarities; DAb-seq is also relevant as an earlier method implementing a DNA/protein co-assay at single cell resolution. I recognize the unique value of what's been developed here for HIV in particular, but this is a pretty minimal ask and a failure to cite or mention the substantial work that's been done in this area is misleading.

We have now integrated citations about CITE-Seq and DAb-Seq into the manuscript and provide more methodological context; many thanks for this comment.